# Spatial extrapolation of stream thermal peaks using heterogeneous time series at a national scale

Aurelien Beaufort[1,2], Jacob S. Diamond[1,2], Eric Sauquet[1,] Florentina Moatar[1*]

[1]RiverLy, INRAE, Centre de Lyon-Grenoble Auvergne-Rhône-Alpes, France 69100
[2]Université de Tours, GéoHydrosystèmes COntinentaux, Tours, France 38000

*Correspondence to*: Florentina Moatar (florentina.moatar@inrae.fr)

**Abstract.** Spatial reconstruction of stream temperature is relevant to water quality standards and fisheries management, yet large, regional scale datasets are rare because data are decentralized and inharmonious. This data discordance is a major limitation for understanding thermal regimes of riverine ecosystems. To overcome this barrier, we first aggregated one of the largest stream temperature databases on record with data from 1700 individual stations over nine years from 2009–2017 (n=45,000,000 hourly measurements) across France (area = 552,000 km$^2$). For each station, we calculated a simple, ecologically relevant metric–the thermal peak–that captures the magnitude of summer thermal maximums. We then used three statistical models to extrapolate the thermal peak to nearly every stream reach in France and Corsica (n=105,800) and compared relative model performances with an air temperature metric. In general, the hottest thermal peaks were found along major rivers, whereas the coldest thermal peaks were found along small rivers with forested riparian zones, strong groundwater inputs, and which were located in mountainous regions. Several key predictors of the thermal peak emerged, including drainage area, mean summer air temperature, minimum monthly specific discharge, and vegetation cover in the riparian zone. Despite differing predictor importance across model structures, we observed strong concordance among models in their spatial distributions of the thermal peak, suggesting its robustness as a useful metric at the regional scale. Finally, air temperature was found to be a poor proxy for the stream temperature thermal peak across nearly all stations and reaches, highlighting the growing need to measure and account for stream temperature in regional ecological studies.

## 1 Introduction

Stream water temperature controls the rates of biogeochemical processes (Nimick et al., 2011; Song et al. 2018) and the distribution and phenology of aquatic communities (Hawkins et al. 1997; Wolter 2007). Water temperature extremes, like summer maxima, are particularly strong drivers of aquatic biota behavior, habitat selection (Carlson et al., 2007; Xu et al., 2010), and water quality (van Vliet and Zwolsman, 2008). Despite their importance, we lack an understanding of the spatial distributions of thermal maximums, largely due to a paucity of consistent stream temperature data. There is thus a need to develop and improve spatially distributed stream temperature datasets with a focus on identifying areas most affected by thermal maximums.

National- and even regional-scale stream temperature data are a relatively recent phenomenon (Jackson et al., 2018), in part due to advancements in *in-situ* sensors (Isaak et al., 2017), but these data tend to be spatiotemporally uncoordinated across regions and river networks (Arismendi et al., 2014). This results

in snapshot datasets, necessitating the use of air temperature proxies to supplement or even replace stream temperature time-series (Conti et al., 2015; Logez et al., 2012; Tisseuil et al., 2012). Indeed, ecologists regularly use air temperature as a proxy or predictor of stream temperature (Buisson and Grenouillet, 2009; Durance and Ormerod, 2009; Mayer 2012) despite the fact that water temperature

better explains the spatial organization of aquatic biota (Kirk and Rahel, 2022). While air temperature can be used to fill temporal data gaps (Morrill et al. 2005), air temperature is a poor surrogate for stream temperature in several cases. In particular, air temperature correlates poorly with stream temperature in headwater reaches (Caissie, 2006), and in reaches with strong local controls (e.g., riparian vegetation, groundwater inflows, bed form, impoundements) (Moatar and Gailhard, 2006; Hill and Hawkins, 2014;

Loicq et al., 2018; Chandesris et al., 2019; Seyedhashemi et al., 2021).

The spatial variation of summer stream temperature is controlled by several factors, each with varying spatiotemporal influence (Fullerton et al., 2015). The most commonly observed factor is stream size–often evaluated with a drainage area or distance-from-source proxy–which tends to have a positive relationship with stream thermal maxima (Hrachowitz et al., 2010; Imholt et al., 2013; Isaak et al., 2017).

As stream's surface area increases, it leads to rapid heat transfer and subsequent equilibration with air temperatures. Intuitively, discharge is also a strong control on stream temperature distributions, with increasing groundwater influence (e.g., measured with monthly flow minima or the base flow index [BFI]) often cited as a mitigating effect on thermal maxima (Chang and Psaris, 2013; Hare et al. 2021). An obvious candidate factor is summer air temperature, which exhibits clear positive relationships with

stream thermal maxima, albeit at regional scales (Moore et al, 2013; Isaak et al., 2017). In addition to these three dominant factors, an array of additional controls like local geology, slope, altitude, and riparian cover can modulate these relationships, leading to unexpected spatial distributions of stream water temperature (Fullerton et al., 2015).

To account for the variety of controls on stream temperature spatial distributions in a data scarce

landscape, researchers turn to predictive models. Still, despite a recent growth in modeling techniques, including statistical (Segura et al. 2015, Benyahya et al. 2007) and physical (Beaufort et al. 2016a,b) approaches, models that can capture regional or national-scale stream temperature distributions remain scarce (but see Hare et al. 2021). At these larger spatial scales, statistical models, which relate stream temperature to climatic and environmental variables, are likely the most pertinent and parsimonious for

management. Indeed, geostatistical river network models that can incorporate network covariance structure are likely the most appropriate and accurate models to estimate spatial distributions of stream temperature (Isaak et al., 2017), but their use is limited to regions with high data density. As most countries lack the required data for such approaches, researchers are more prone to employ empirical regressions or machine learning approaches, but the comparative accuracy and utility of these

approaches is still unknown.

In light of these gaps in data and understanding, our objectives were twofold: 1) to aggregate and develop a homogeneous national-scale stream temperature database for France, and 2) to develop empirical

statistical models to predict a simple, ecologically relevant stream thermal metric that captured the magnitude of the stream temperature maxima at the regional scale. To do so, we first defined an

interannual thermal metric, which we termed the "thermal peak", using heterogeneous and non-concomitant time series of stream temperature and estimated it at 1700 stations for 2009–2017. We then tested three statistical models and one multi-model approach to predict this metric at the regional scale and compare their predictive capacity with that of air temperature. Specifically, we used these models to address the following question: what are the spatial patterns of stream temperature maxima in France

and their drivers, and are these patterns consistent across modeling approaches? We hypothesized that spatial patterns would be consistent, whereas the drivers would depend on the modeling approach used. We also hypothesized that stream size, air temperature, and groundwater contributions would emerge as important, regardless of approach.

## 2. Methods

### 2.1 Study area and monitoring network

The study area is continental France and Corsica (550,000 km$^2$). France is located in a temperate zone characterized by a variety of climates due to the influences of the Atlantic Ocean, the Mediterranean Sea, and mountain areas. We assembled a large stream temperature dataset for this area by combining data from both public (national, regional; approximately 600 stations available from http://www.naiades.eaufrance.fr/), fishing associations and

other private and public sources sub-national agencies. Due to the diversity of data origins, the assembled time series did not have a consistent spatial and temporal structure. Some regions are densely monitored while others had few instrumented streams and much of the data do not exhibit temporal overlap, challenging our ability to define comparable metrics among streams. Hence, our main challenge was to coalesce and homogenize all the disparate data sources.

The stream temperature data used here comprise approximately 45,000,000 hourly measurements from 1,700 unique measurement stations (n=2,107,623 site-days) collected between 2009 and 2017, primarily during summer. All the stations under strong human influence (i.e., dam releases and nuclear power thermal effluent) were excluded from this data set. All data were recorded by automatic data loggers managed by professional biologists, hydrologists, and fishermen. Outliers from each station's time series were removed with automatic outlier

detection filters and the resulting hourly data were screened visually before being averaged into daily mean stream temperature data (hereafter referred to as $T_w$). The automatic outlier detection consists of three steps with eight unique filters that remove, in order, 1) stream temperature anomalies based on hourly data, 2) anomalies between $T_w$ and daily mean air temperature ($T_{air}$) at monthly scales, and 3) anomalies between $T_w$ and $T_{air}$ at daily scales (Table 1; Moatar et al., 2001; Beaufort et al., 2020b). $T_{air}$ was provided by the 8 km gridded SAFRAN (Système

d'Analyse Fournissant des Renseignements Atmosphériques à la Neige) atmospheric reanalysis data released by Meteo-France over 2009-2017 period (Vidal et al., 2010). It was extracted from SAFRAN meshes overlapping the station location.

**Table 1.** Outlier detection and data filtering process of stream temperature dataset

| Filter type | Definition | Threshold (°C) | Data removed (%) |
|---|---|---|---|
| Stream temperature anomalies from hourly data | Maximum temperature > threshold (by month) | 14, 15, 20, 24, 28, 30, 32, 33, 29, 28, 18, 17 (Jan.–Dec.) | 2 |
| | Minimum temperature < threshold | -0.5 | |
| | Difference in consecutive data > threshold | 2 | |
| | Daily diel range > threshold | 7 | |
| | Difference in daily max. or min. in consecutive data > threshold | 3 | |
| Monthly $T_w$–$T_{air}$ anomalies | $R^2$ of daily regressions by month < threshold | 0.1 [unitless] | |
| | Deviation of the monthly difference ($T_w$ –$T_{air}$) to its interannual mean > threshold | 4 | |
| Daily $T_w$–$T_{air}$ anomalies | Daily difference ($T_w$ –$T_{air}$) > the monthly mean of daily differences by threshold | 2.5 | 5 |

To understand spatial patterns in ecologically meaningful temperature metrics, we focused on the two hottest stream temperature months, July and August (hereafter referred to as summer), a time period essential for the growth and survival of many aquatic species. This focus also had the benefit of maximizing the number of observation stations for analysis. Still, out of the 1700 stations, 490 stations had just one year of data, 88 stations had observations covering summer over all nine years, and only 30 had year-round observations for all nine years (Figure 1a,b). To obtain hydraulic and hydrologic characteristics for each station, stations were projected onto the Theoretical Hydrographic Network for France (RHT; Pella et al., 2012), an oriented hydrographic network with defined flow directions (i.e., built-in upstream-downstream dependencies) that comprises 114,600 reaches of median length 1,961 m (2,475±1,512 m, mean±sd). A majority of stations were located on RHT river reaches with drainage areas 20–500 km², whereas most reaches are small streams with a drainage area of less than 20 km² (Figure 1c).

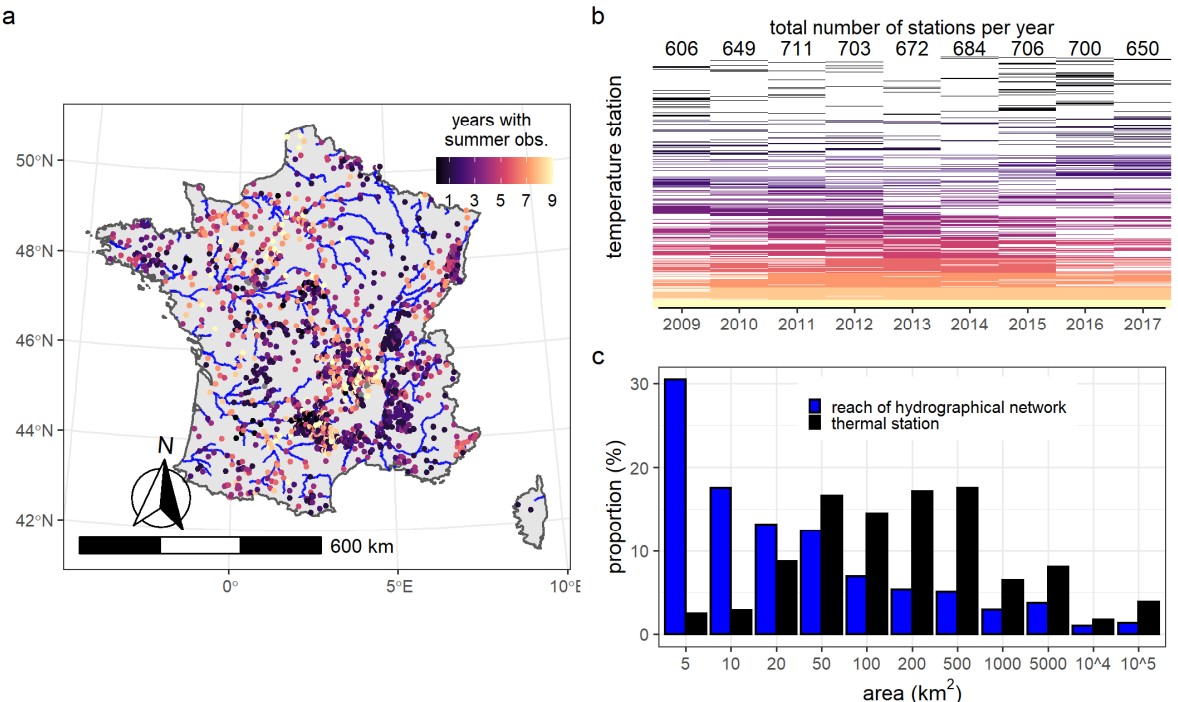

**Figure 1.** Data availability for each temperature station used in this study. a) Map of stream temperature stations in France with RHT network shown for all reaches of Strahler order >4, b) Heatmap of data availability by year (x-axis) and station (y-axis) with the total stations per year listed at the top of each column. Sites are colored by the number of years with observations, and c) Distributions of drainage area for RHT reaches (blue) and of thermal stations (black).

## 2.2 Defining the thermal peak metric

Due to the limited concordance among stream temperature time series (Figure 1b) and to focus our analysis towards ecologically relevant ends, we summarized stream temperature data with a simple metric, the thermal peak. This metric has precedent in regional species distribution models that instead used air temperature (hereafter referred to as $T_{air}$) as a proxy (Buisson and Grenouillet, 2009). We refer to this metric as the thermal peak ($T_p$), and define it as the interannual average of the mean temperature of the 30 hottest consecutive days of each year $\left(\overline{T_{w,30}}\right)_i$:

$$T_p = \frac{\sum_{i=1}^{N}\left(\overline{T_{w,30}}\right)_i}{N} \tag{1}$$

where:

        i = year index; and

        N= the number of years of observation available from 2009–2017 (N = 1–9)

We did not know *a priori* the 30 hottest consecutive days of each year, but a sensitivity analysis on the sites with annual data suggests that July and August were regularly the hottest months (Fig: B1). Indeed, across sites with annual data, the hottest day of the year always occurred within the approximate 30 day period between July 28 and August 30 (mean±sd: August 12±16 days), and only 3% of site-years had their hottest 30-days outside of this period. This fact further allowed us to take advantage of many of the sub-annual time series, particularly those generated by fishing agencies, which only contain July and August data.

## 2.3 Climate correction of the thermal peak

The thermal peak can be biased depending on the climatic variability of the years of observation for each station. Indeed, only 30 of our stations had $T_p$ calculated using all nine years of data, and therefore had the highest level of confidence in their estimate. We refer to the $T_p$ from these 30 stations as $T_{p,ref}$, indicating that these are reference, or true estimates of $T_p$ (Table 2). To account for the bias associated with missing data at the remaining 1670 stations, we gap-filled missing data at these stations using site-specific stream-air temperature regressions. This method accounts for interannual variation in climatic forcing on stream temperature, and we therefore refer to it as a climate correction.

The climate correction is achieved by first calculating station-specific regressions during summer between daily $T_w$ and a right-aligned moving average of $T_{air}$ at lags ranging from 2–10 days. The moving average lag whose regression produced the highest coefficient of determination was then used to fill gaps in the time series of $T_w$ for each station. Stations with watershed areas greater than 1000 km$^2$ tended to have the best R$^2$ at the longest lags, but there were no clear trends at smaller watershed areas (Fig. B2). Next, we reconstructed summer stream temperature using this regression and subsequently recalculated $T_p$ based on this reconstructed data. We refer to $T_p$ from these climate-corrected data as $T_{p,clim}$ to indicate that missing data were gap-filled with the climate correction procedure (Table 2).

To validate this approach, we conducted a permutation test on the 30 stations with full annual monitoring from 2009–2017 (i.e., sites where a $T_{p,ref}$ is known). At each site, we introduced randomly placed, artificial annual gaps into observed data ranging from 1–9 years to simulate missing data. We then backfilled these introduced gaps according to the climate correction method. Following gap-filling, we calculated two metrics: 1) $T_p$ using gap-induced data without gap-filling ($T_{p,gap}$), and 2) the thermal peak using the gap-filled data ($T_{p,fill}$; Table 2). We then compare these two metrics to the reference thermal peak ($T_{p,ref}$) using absolute biases at each tested permutation (i.e., the number of introduced gap years). This approach allowed us to assess whether the climate-corrected reconstruction of the gaps in time series is 1) a useful approach, and 2) lower in bias and uncertainty compared to using observed data alone.

**Table 2.** List of thermal peak terminology with the count of sites (n) used in their calculation

| Notation | Definition | n |
|---|---|---|
| $T_{p,obs}$ | thermal peak, or the interannual average of the mean temperature of the 30 hottest consecutive days of each year for station with observations | 1700 |
| $T_{p,ref}$ | thermal peak for reference stations with all nine years of data | 30 |
| $T_{p,clim}$ | thermal peak for stations with less than nine years of data to which climate correction was applied | 1670 |
| $T_{p,gap}$ | thermal peak for reference stations with introduced data gaps | 30 |
| $T_{p,fill}$ | thermal peak for reference stations whose introduced data gaps were filled with climate correction | 30 |
| $T_{p,m}$ | modeled thermal peak using statistical extrapolation to the RHT network using statistical method m | 114,600 |
| $T_{p,air}$ | thermal peak estimated using the SAFRAN reanalysis $T_{air}$ data | 114,600 |

**2.4 Extrapolating the thermal peak to national scale with statistical modeling**

We estimated $T_p$ throughout the entire RHT network using four distinct statistical models. For modeling ($T_{p,m}$) it was not clear how to choose *a priori* a particular model structure due to the complexity of the processes involved in determining local stream temperature. Therefore, we tested four different structures: 1) a multiple linear regression model (REG), 2) an artificial neural network model (ANN) that is potentially non-linear, but encompasses a linear model as a special case, 3) a random forest model (RF) that can also be non-linear, but with a different strategy than ANN, and 4) a multi-model combination (MM), which combines the three prior structures ANN, RF, and REG. All these models are based on a function to estimate $T_p$ at all stations (i):

$$T_{p,m,i} = f(g_{1,i}, \dots, g_{n,i}) \tag{5}$$

Where $g_{n,i}$ = the nth explanatory variable defined for each ith station.

To measure how stream thermal regime estimations might be different if using a $T_{air}$ proxy, we also calculated $T_{p,air}$ using $T_{air}$. The $T_{p,air}$ is calculated identically to $T_p$, but using daily $T_{air}$ instead of daily $T_w$, as would be done in ecological studies using $T_{air}$ as a proxy for $T_w$. Finally, to determine the effect of climate corrections on the full RHT extrapolation, we compared the distribution of $T_{p,m}$ values when models were fitted using either $T_{p,obs}$ or $T_{p,clim}$.

**2.4.1 Explanatory variables**

We selected sixteen variables (Table 3) to explain the spatial distribution of $T_{p,m}$ based both on results from a prior analysis (Beaufort et al., 2020a), review of the literature and an effort to minimize variable collinearity (Fig. B3). We further considered the ability to calculate or estimate each variable at the scale of the entire RHT network. The variables fall into three categories: climatic, hydrologic, and watershed characteristics.

The four climatic variables were determined from SAFRAN reanalysis data for the years 2009–2017: 1) mean annual precipitation, 2) mean summer precipitation, 3) mean annual snowfall, and 4) mean summer air temperature.

The four hydrological variables were determined by extrapolation based on prior datasets. In general, we wanted to quantify the influence of groundwater on $T_p$, but lacked the local hydrometric data necessary to calculate the typically used BFI (Hare et al. 2021) at a majority of sites. Hence, we turned to the following variables that could potentially quantify groundwater influence across all of our sites. The first two variables, monthly minimum discharge (Sauquet et al. 2008) and the annual minimum monthly discharge with a return period of five years

(Catalogne, 2012), describe the low-flow regime of each site. The remaining two variables, the hydrologic regime (HR; Sauquet et al., 2008) and the concavity index (CI; Catalogne, 2012), are dimensionless and characterize the general hydrology of each site. More specifically, the HR groups sites into one of 12 classes ranging from rainfall-dominated, to transitional, to glacial and snow melt dominated (see Appendix A). These classes generally fall into buffered (i.e., high baseflow) or highly variable (i.e., low baseflow) hydrologic regimes. The CI describes the

concavity of the flow duration curve, where values close to 1 indicate low flow variability (e.g., large high storage capacity in aquifer or snow) and values close to 0 indicate high flow variability (e.g., low storage capacity exemplary of Mediterranean systems). The concavity index is computed as follows:

$$IC = \frac{Q_1 - Q_{10}}{Q_{10} - Q_{99}} \tag{6}$$

where $Q_p$ is the daily flow exceed p% of the time derived from the flow duration curve.

The eight variables relating to watershed characteristics were extracted from either the SYRAH-CE (SYstème Relationnel d'Audit de l'Hydromorphologie des Cours d'Eau) database (Valette et al., 2012) or the RHT database (Pella et al., 2012). The three variables from RHT comprise: 1) mean altitude, 2) watershed drainage area, and 3) mean slope. The five variables from SYRAH-CE comprise: 1) riparian vegetation cover in a 10 m buffer, 2) linear upstream weir density along the stream, 3) areal upstream weir density for the watershed, 4) upstream pond cover

as a fraction of stream area, and 5) incision class describing the rate of incision of the valley.

**Table 3.** List of explanatory variables used in models and hypothesized effects on thermal peaks

| Category | Variable [units] | Notation | Source | Hypothesized effect | Reference |
|---|---|---|---|---|---|
| Climatic[a] | Mean annual precipitation [mm] | $P_{annual}$ | SAFRAN | Reduced $T_p$ via increased baseflow | Strauch et al. 2017 |
| | Mean summer[b] precipitation [mm] | $P_{summer}$ | SAFRAN | Increased summer $T_p$ via warm runoff | Nelson and Palmer, 2007 |
| | Mean annual snow accumulation [mm] | $S_{annual}$ | SAFRAN | Reduced $T_p$ via colder snowmelt water | Caissie, 2006; Webb et al., 2008 |
| | Mean summer[b] air temperature [°C] | $T_{summer}$ | SAFRAN | Increased summer $T_p$ | Moore et al, 2013; Isaak et al., 2017 |
| Hydrologic | Mean annual specific discharges [L s$^{-1}$ km$^{-2}$] | $Q_{mean}$ | RHT | Reduced $T_p$ via greater thermal capacity | Caissie, 2006 |
| | Mean monthly minimum specific discharge* [L s$^{-1}$ km$^{-2}$] | $q_{min}$ | RHT | Increased $T_p$ via lower baseflow | Chang and Psaris, 2013 |
| | Concavity index† [-] | CI | RHT | Reduced $T_p$ at higher CI | This paper |

| | | | | | |
|---|---|---|---|---|---|
| | Hydrological regime‡ [-] | HR | RHT | Reduced $T_p$ under high baseflow regimes and vice versa | This paper |
| Watershed characteristics | Mean watershed elevation [m] | elev | RHT | Reduced $T_p$ via low $T_{air}$ | Isaak and Hubert, 2001 |
| | Drainage area [km²] | area | RHT | Increased $T_p$ due to greater thermal exchange | Hrachowitz et al., 2010; Imholt et al., 2013 ; Isaak et al., 2017 |
| | Mean slope of the watershed [m km⁻¹] | slope | RHT | Reduced $T_p$ via reduced insolation exposure time | Daigle et al., 2010 |
| | Riparian vegetation cover ratio in 10 meters buffer (%)** | veg | SYRAH | Reduced $T_p$ via shading | Moore et al., 2005 |
| | Linear weir density upstream of stations (# km⁻¹)** | weirs | SYRAH | Increased $T_p$ via warming | Chandesris et al., 2019 |
| | Areal weir density upstream of stations (# km⁻²)** | weir area | SYRAH | Increased $T_p$ via warming | Chandesris et al., 2019 |
| | Pond cover ratio upstream of stations (%)** | ponds | SYRAH | Increased $T_p$ via warming | Seyedhashemi et al., 2021 |
| | Stream incision class ** | SI | SYRAH | Reduced $T_p$ for greater incision | Webb et al., 2008 |

[a]all climate variables are calculated on data from 2009–2017
[b]summer refers to July–August
*determined by geostatistical interpolation on the RHT network (Sauquet et al., 2000) with return period of five years
†ratio of flow duration quantiles $Q_1$-$Q_{10}$/$Q_{10}$-$Q_{99}$, as defined by Catalogne (2012)
‡classes from 1–12 with pluvial regimes from 1–6, transition regimes from 7–8; and glacial and snow melting regimes from 9–12 (Sauquet et al., 2008)
**Detailed description of these variables in Valette et al. (2012).

**2.4.2 Multiple regression**

We first fit a multiple linear regression model between $T_p$ and explanatory variables using all possible variables characterized in Table 3. Prior to fitting, we scaled the explanatory variables so that their fitted coefficients could be compared in terms of relative importance. We did not use any variable selection techniques in the multiple regression approach because our goal was to compare across the four modeling approaches (regression, ANN, random forest, and multi-model) that use the same independent variables. In other words, we did not seek to have the most parsimonious multiple regression model, but instead used all 16 variables (Table 3) to create the model. Each of these variables was previously assessed for multicollinearity (Fig. B3), and variance inflation factors were all less than 3.

**2.4.3 Artificial neural network**

We then used an ANN—specifically a feed-forward neural network with one hidden layer (R package *nnet*; Venables and Ripley 2002)—to estimate $T_p$ as a potentially non-linear function of covariates. An ANN is a flexible mathematical structure with an interconnected set of simple processing elements, called nodes or neurons, that emulates the functioning of neurons in the human brain. They are used as an alternative tools to identify and simulate complex and highly nonlinear relationships between input and output data that ANN identifies during the 'learning process'. We included a direct connection between covariate inputs and outputs so that the case with zero hidden units corresponded to a linear relationship. We used weight decay regularization, also known as ridge regression, to control overfitting by decreasing less relevant coefficients. Both the number of hidden units and the amount of weight decay were selected with a first cross-validation procedure (Bishop, 2006). To quantify the importance of the different covariates, we used a connection weight approach (Olden and Jackson, 2002; Olden et al., 2004):

$$W_V = \sum_{h=1}^{nhu} A_{V,h} B_h \qquad (7)$$

where

$W_V$ (-) = the relevance of covariate V,

$A_{V,h}$ (-) = the ANN coefficients connecting hidden unit h to covariate V,

$B_h$ (-) = the ANN coefficients connecting hidden unit *h* to the output, and

nhu = the number of hidden units

### 2.4.4 Random forest

We used a random forest (RF) for the third statistical model structure. RFs is a machine-learning method that draws bootstrap samples from original training data and fits a tree to each sample and the ensemble of trees are used for regression (Breiman, 2001). Each tree recursively partitions the individuals from each sample into non-overlapping groups of each predictor, which are then used to predict the target variable, in this case $T_p$. The average across the ensemble of trees is used as the final regression. The importance of each predictor variable was provided

as standard output by the *randomForest* package, which determines how much the mean square errors in prediction increases when that covariate is randomly permuted within the tree. We used the R package "randomForest" (Liaw & Wiener, 2002), following Breiman's algorithm (Breiman, 2001) allowing 500 trees.

### 2.4.5 Multi-model combination

Finally, we used a multi-model combination approach to obtain a consensus estimation map of $T_{p,m}$. The

temperature predictions from each previously described model were linearly combined to reduce the associated uncertainties through multiple linear regression.

$$T_{p,mm,i} = a T_{p,ANN,i} + b T_{p,RF,i} + c T_{p,REG,i} + d \qquad (8)$$

where:

$T_{p,mm,i}$=multi-model $T_p$ based on observed and reconstructed $T_{w,i}$ for each observation station, i;

$T_{p,ANN,i}$=$T_p$ estimated by ANN for each observation station i;

$T_{p,RF,i}$=$T_p$ estimated by RF at each observation station i;

$T_{p,REG,i}$=$T_p$ estimated by multiple regression at each observation station i; and

a, b, c, d=fitted regression coefficients.

Temperature predictions made by the three models are used as multi-model independent variables and each model

prediction is weighted with a coefficient to match the observations as closely as possible. Hence, the multi-model coefficients are calculated only in relation to observations at the 1700 stations. Then, using equation (8) with calculated coefficients, we extrapolate $T_p$ along the river network.

### 2.4.6 Model cross-validation and comparison with air temperature

To assess each model's performance prior to spatial extrapolation to the entire RHT network, we conducted a

cross-validation on observed data from our 1700 stations. For each model, including $T_{p,air}$, we used 80% of stations (n = 1360) as training data to estimate model parameters. Using those model parameters, we estimated validation data $T_p$ at the remaining 20% of stations (n=340) and cross-validated those estimates with $T_{p,obs}$. We conducted this cross-validation 100 times allowing for random selection of stations used in the training and validation data sets. We evaluated the results of the cross-validation with the Nash-Sutclife efficiency criterion (NSE, Nash and

Sutcliffe, 1970), the RMSE (Root Mean Square Error) and biases between observed $T_{p,obs}$ and $T_{p,m}$. We also conducted regressions of $T_{p,obs}$ and $T_{p,m}$.

We also compared all $T_p$ model results with a thermal peak calculated using only $T_{air}$ from SAFRAN reanalysis data, $T_{p,air}$. The goal was to evaluate model performance with the simplest and most widely available stream temperature proxy.

**2.4.7 Hypothesis testing**

We evaluated the hypothesis that spatial patterns across models would be consistent, but that drivers would depend on the modeling approach in two ways. First, we compared relative importance of the most important variables across models to see if the same variables emerged. Second, we visually compared maps of $T_{p,m}$ and compared their distributions across watershed area and stream order. We evaluated our second hypothesis that stream size,

air temperature, and groundwater contributions would emerge as important regardless of approach by comparing their relationships with $T_{p,m}$ across modeling approaches using Kendall tau correlation.

**3. Results**

**3.1 Validation of the climate correction approach**

Gap-filling $T_w$ with our climate correction approach resulted in consistently lower absolute biases for $T_p$ (i.e., $T_{p,fill}$)

compared to only using available data (i.e., $T_{p,gap}$), regardless of the length of introduced annual gaps (Figure 2). This discrepancy in bias between uncorrected and corrected data increased exponentially with the number of introduced gaps, growing from a median of 0.03°C with one year of gaps to 0.25°C with eight years of gaps. Climate corrected biases remained below 0.25°C for 75% of the stations with up to six years of introduced gaps. There was no systematic bias regardless of the years taken into account in the regressions.

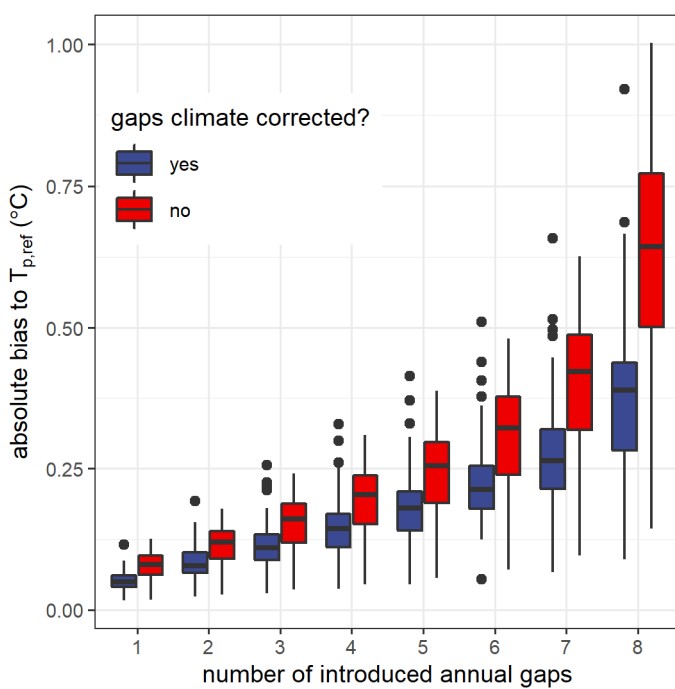


**Figure 2.** Improvement in absolute bias of thermal peaks across all reference sites ($T_{p,ref}$; n=30) when introduced annual gaps were filled with the climate correction procedure ($|T_{p,fill}–T_{p,ref}|$, red boxplots) compared to when gaps were unfilled ($|T_{p,gap}–T_{p,ref}|$, blue boxplots) regardless of the number of introduced gaps.

### 3.2 Model cross-validation

In cross validation, all four statistical models performed substantially better than air temperature at accurately predicting $T_p$ (Figure 3). Indeed, $T_{p,air}$ overestimates the observed $T_{p,ref}$ by 2.5°C on average (Figure 3a), and the negative NSE for $T_{p,air}$ indicates that using a simple mean of $T_w$ observations is a better predictor of $T_p$ than using $T_{air}$ (Figure 3b). Overall, models tend to slightly underestimate $T_p$ (Figure 3a), but mean biases are close to 0°C. The REG and ANN models had similar performance (median RMSE > 1.5°C; median NSE = 0.6), whereas the RF and MM models obtained the best performances (median RMSE < 1.5°C; median NSE = 0.7) with the MM slightly superior for NSE (Figure 3b,c). The models explain in cross-validation between 70% and 78% of the variation in $T_{p,obs}$ with the best performances for RF and MM (Figure 3d-f). Sites with larger watershed areas had consistent positive bias (lighter colors in Figure 3d-g), but bias was more evenly distributed for sites with smaller watershed areas, thought the smallest watersheds tended to exhibit negative bias. We did not observe any strong spatial patterning in performance metrics, although sites with larger watershed areas tended to have positive bias (Figure 3d–h).

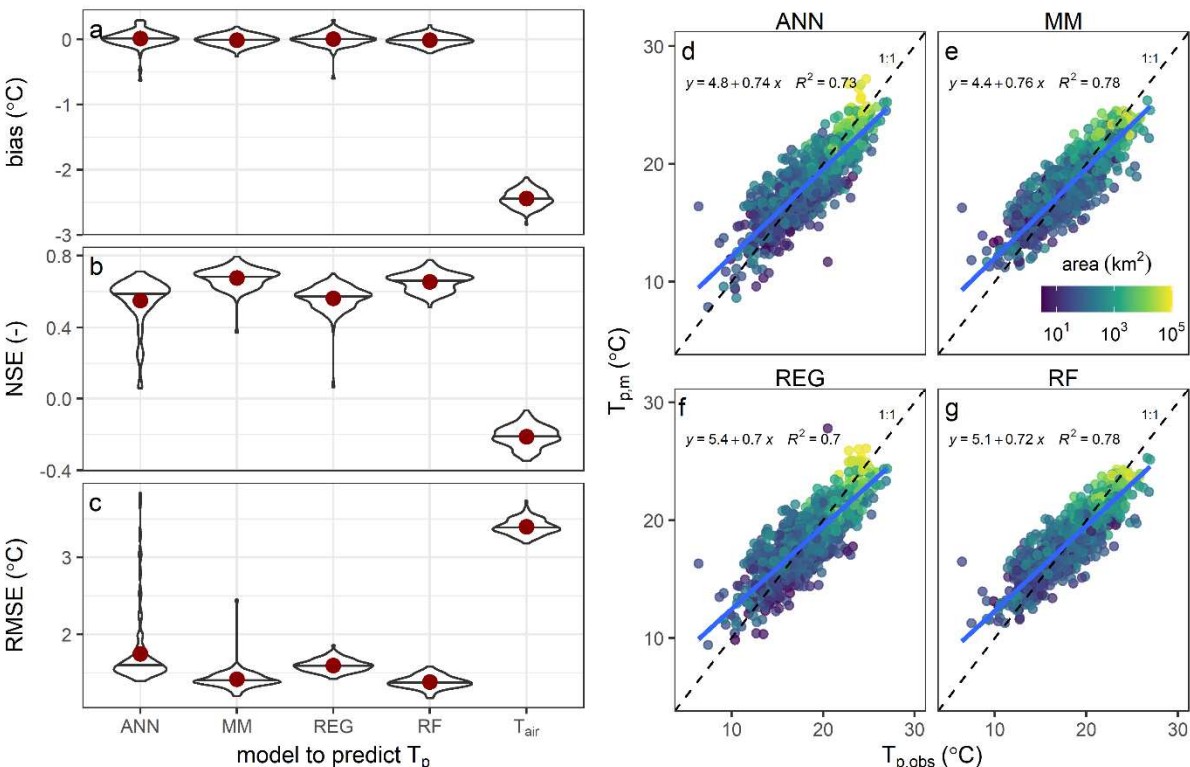

**Figure 3.** Cross validation performance metrics for modeled thermal peaks ($T_{p,m}$), where 80% of stations (n = 1360) were training data and 20% of stations (n=340) were validation data. Violin plots show distributions of 100 replicates of training-validation data for each model's performance metrics: a) bias relative to $T_{p,ref}$; b) NSE; c) RMSE of $T_{p,m}$ relative to observed $T_p$. Horizontal lines indicate sample medians and red points indicate sample means. d-f) $T_{p,m}$ vs. $T_{p,obs}$ for each model colored by watershed area at the site. Simple linear regression results (n = 1700) are shown in blue with comparison to a 1:1 line (dashed).

### 3.3 Explanatory variables importance and effects in models

Variable importance differed among modeling approaches, and the maximum importance value for any one variable was between 25–30% (Figure 4). The two most important variables were watershed area (area) and mean summer air temperature ($T_{summer}$) for the RF and multiple regression models (Figure 4b,c). For the ANN model, minimum monthly specific discharge ($q_{min}$) and riparian vegetation cover (veg) were most important (Figure 4a; note that $q_{min}$ is not correlated with area, Figure B3). Surprisingly, area and $T_{summer}$ obtained relative importances

of less than 5% in ANN whereas they were the most important variables in the RF and the REG models. Across models, none of the three precipitation variables (Table 3) emerged as important, nor did the stream incision class. The cumulative importance of the four most relevant variables of the REG and ANN models is respectively 92% and 81%, which means that the other variables had very little weight in the estimates. This sum is only 69% for RF, which indicates that the relative importance of the explanatory variables are more distributed and the other

explanatory variables had a significant weight in the estimates which could explain the best performances in cross validation of RF. In particular, linear and areal weir density upstream of stations occupied a cumulative 7% relative importance. These variables also occupied the two next greatest importance rankings (at 1.7% and 0.9% relative importance, respectively) for the multiple regression model after those shown in Figure 4.

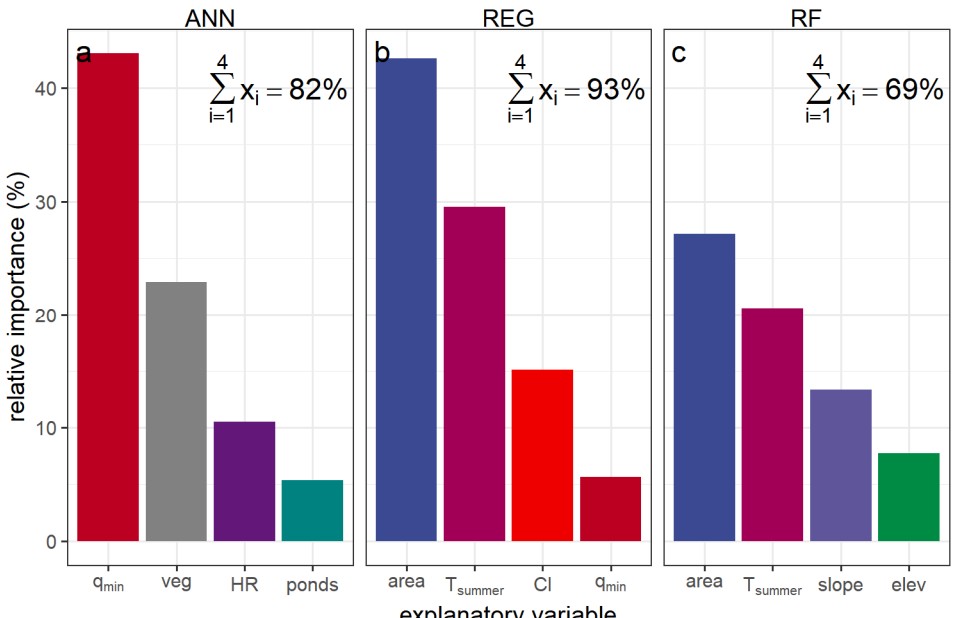

**Figure 4.** Relative importance of top four explanatory variables calculated in cross validation for the estimation of $T_p$ with the models: a) ANN; b) REG and c) RF. To compare the relative importance of each variable for each model, all variables were centered and scaled. Text in the upper right of each panel refers to the sum of the relative importances of the first four explanatory variables in each model; colors indicate the explanatory variable.

The direction and magnitude of effects from the hypothesized variables (area, groundwater contributions, and summer air temperature) did not systematically differ across modelling approaches (Figure 5). However, there were minor differences in that the RF model tended towards higher $T_{p,m}$ than ANN and REG models at small watershed areas and tended towards lower $T_{p,m}$ than ANN and REG models at larger watershed areas (compare Fig. 5g with 5a and 5d). The RF model was also less sensitive to $T_{summer}$, with RF producing lower $T_{p,m}$ than ANN

and REG models at sites with cooler summer temperatures and lower $T_{p,m}$ than ANN and REG models at sites
        with warmer summer temperatures (compare Fig. 5i with 5c and 5f).

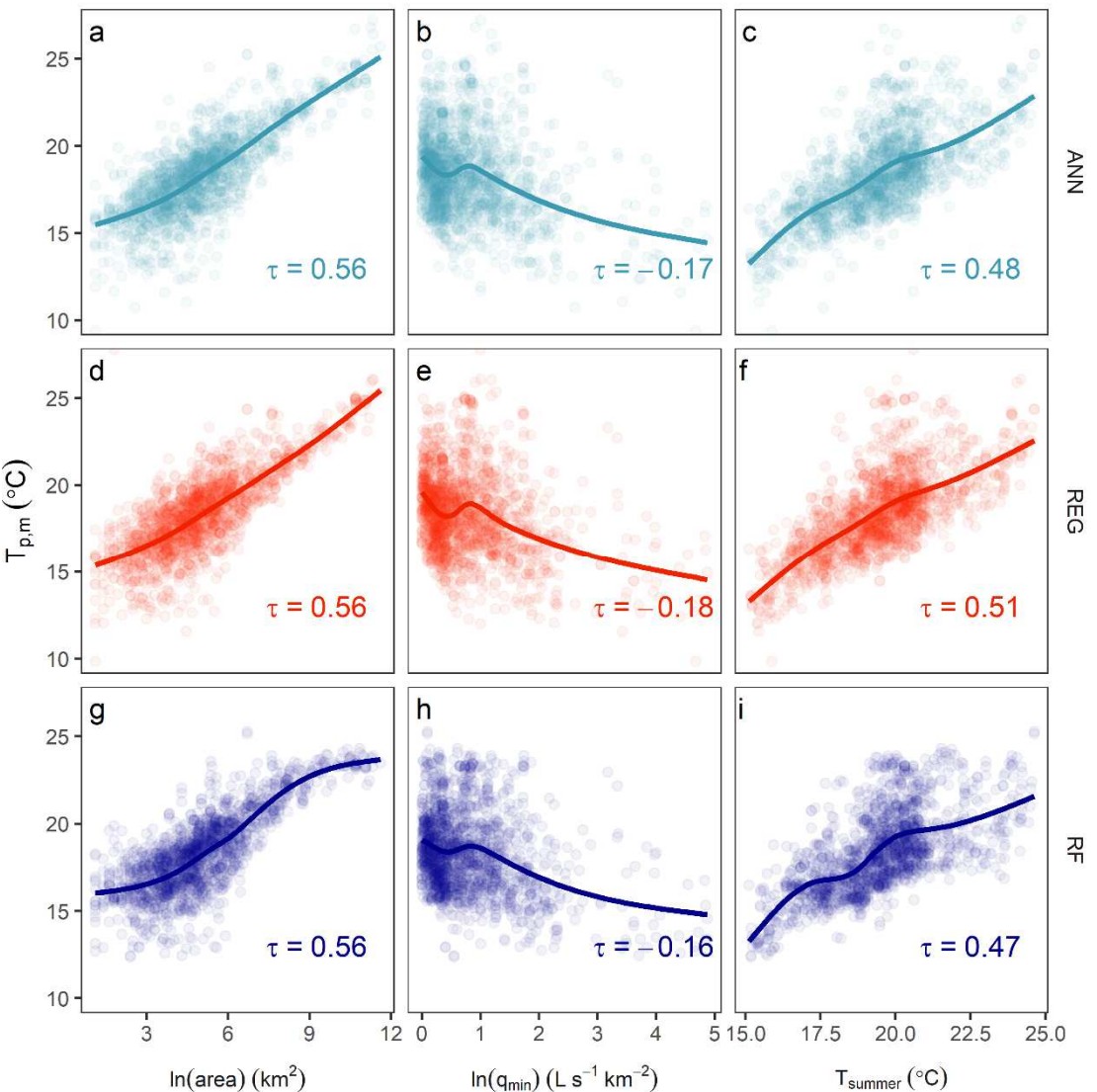

**Figure 5.** Modeled $T_p$ at each of the 1700 observation stations as a function of the variables hypothesized to be
equally important across approaches. Rows indicate the model used (points also colored accordingly) and columns
are separated by area, $q_{min}$, and $T_{summer}$; area and $q_{min}$ are natural-log transformed. Lines indicate best-fit smoothers
        using a generalized additive model and text indicates Kendall tau correlation values (all p<0.001). There was no
        systematic difference in variable effects on $T_{p,m}$.

### 3.4 Spatial extrapolation of thermal peaks and comparison with air temperature

The statistical models could extrapolate $T_p$ to 92% (105,800 reaches) of the RHT network, and the resulting spatial
structure of the extrapolations was consistent across models (Figure 6). On average, $T_p$ was 18.2°C and ranged
        between 6.3°C and 27.0°C. The highest $T_{p,m}$ (i.e., $T_p$ > 22°C) were generally found on the largest rivers located in
        the southeast and in the sedimentary plains. The lowest $T_{p,m}$ are found in the mountain streams of the Alps,
        Pyrenees and Massif Central, and in the northwest. Although the distribution $T_{p,m}$ is consistent among models
        (Figure 6), there are some clear disparities, particularly at the extremes. The ANN model predicts lower $T_{p,m}$ (20%

of reaches $T_{p,m}<15°C$) compared to the RF and multi-model (< 10%; Fig B4a). Application of the climate correction prior to model fitting and subsequent extrapolation showed that differences at the RHT are less than 0.5°C at 89% of the reaches (Fig B4b). Still, 10% of the reaches exhibited $T_{p,m}$ differences greater than 0.5°C, and the vast majority of these differences are negative (9.4% vs. 1.6%), suggesting that climate correction more often than not reduces overestimates in $T_p$.

Estimating $T_p$ with air temperature (i.e., $T_{p,air}$) led to consistently higher values than were obtained with statistical models, with $T_{p,air}$ greater than 20°C for more than 70% of the reaches (Figure 6a, Fig B4a). Indeed, depending on the model, 94–97% of reaches had $T_{p,m}$ lower than $T_{p,air}$ (Figure 6) and more than 95% of these were in reaches with watershed areas less than 1000 km². $T_{p,air}$ exhibited its greatest overestimations in the smallest watersheds (Figure 7a) and switched from overestimation to underestimation relative to $T_{p,m}$ at reaches with watershed areas

between 2000–5000 km². While this behavior was similar across models, the REG and ANN models tended to be produce lower $T_{p,m}$ for small areas and higher $T_{p,m}$ for large areas compared to RF and MM models, suggesting their increased sensitivity to this variable. $T_{p,air}$ also increased its overestimations relative to $T_{p,m}$ in reaches with the most groundwater contributions (Figure 7b). Similar to watershed area, the REG and ANN models tended to be produce higher $T_{p,m}$ for low $q_{min}$ and lower $T_{p,m}$ for high $q_{min}$ than the RF and MM models, suggesting higher

sensitivity to this variable. In other words, in small watersheds with low baseflow, $T_{p,m}$ from REG and ANN models were 2–3°C less than from RF and MM models, but in large watersheds with high baseflow, $T_{p,m}$ from REG and ANN models were 3–5°C greater than from RF and MM models.

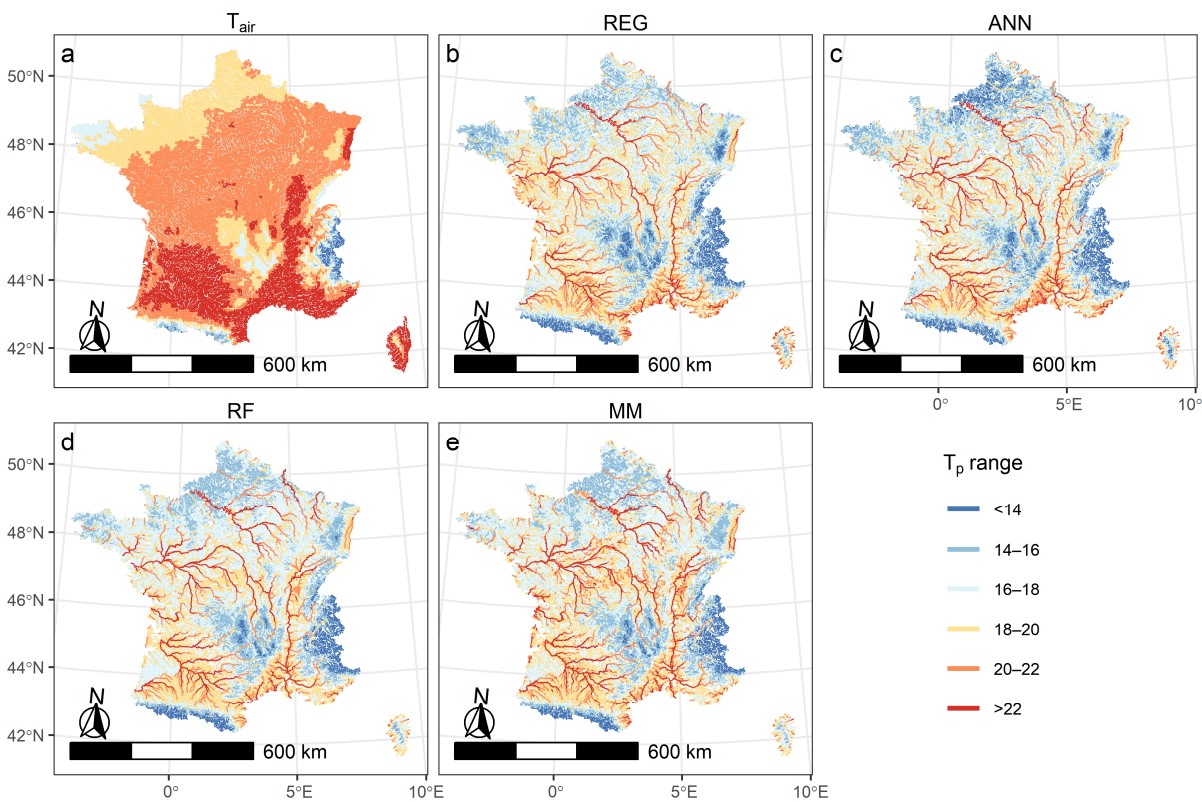

**Figure 6.** Thermal peaks ($T_p$) of stream water extrapolated to all reaches of the French hydrographical network

RHT with the different predictive model structures: a) air temperature ($T_{air}$), b) multiple regression (REG), c)

artificial neural network (ANN), d) random forest (RF), and e) multi-model combination of all previous models (MM). All reaches are colored by their modeled range of $T_p$ and colors are chosen to improve visualization.

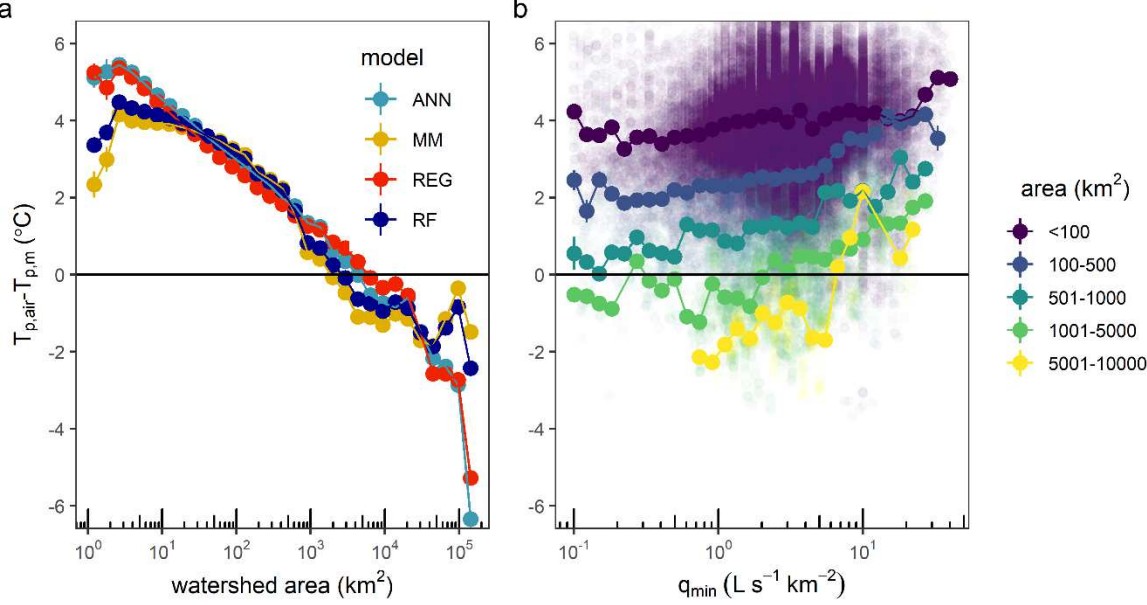

**Figure 7.** Differences between stream thermal peaks estimated by air temperature and models depending on a) watershed area, and b) minimum monthly specific discharge. To simplify visualization across the 114,600 reaches, points were summarized across 5% bins in the x-axis; points indicate mean values at each bin, with vertical error bars indicating standard error. a) As watershed area increases, $T_{air}$ shifts from overestimation to underestimation relative to $T_{p,m}$, with a shift in sign between 2,000 and 5,000 km$^2$. b) As minimum monthly specific discharge increases, $T_{air}$ tends to increase its overestimation relative to $T_{p,m}$, especially for watershed size greater than 100 km² (colors). Only results for RF model are shown for clarity of presentation and because it had the best performance (each model had similar patterns).

## 4. Discussion

We compiled one of the largest regional summertime stream temperature datasets to address the growing need to understand the spatial distributions of stream thermal maxima. This database is available for use at the following website: https://thermie-rivieres.inrae.fr. Using this database, we demonstrated that a simple, ecologically meaningful metric, which we term the thermal peak ($T_p$), can be reliably estimated at the regional scale using a few easily accessible explanatory variables.

### 4.1 Horizons and limitations in estimating large-scale stream thermal metrics

Spatiotemporally comprehensive stream temperature datasets are rare because interest in these data is relatively recent and there is little money to support instrumentation at regional or national scales. This lack of centralized data has been recognized as a major limitation for understanding thermal regimes of riverine ecosystems (Arismendi et al., 2012; Ouellet et al., 2020). Here, we alleviated these limitations by homogenizing existing disparate datasets and demonstrated the aggregated utility of such datasets on improving our understanding of stream temperature distributions. Indeed, our homogenized dataset allowed us to identify, at a national scale, a

map of summer stream temperature maxima with important implications for aquatic species distributions under climate change. We note here that the more detailed computation of $T_p$ (i.e., rolling window of 30-days) could be simplified with a mean of August stream temperature data, as we observed this to regularly be the hottest month across all observable data.

We observed the hottest $T_p$ along major rivers and the coldest $T_p$ along small rivers and in mountainous regions (Fig. 6). The downside of the current approach is that is based on interannual metrics. Indeed, the non-concomitance of the time series precluded us from comparing extreme years (hot vs. cold). However, the stream temperature dataset used here contains enough interannual information that it could be used to calibrate models and assess the impact of climate change on thermal regimes (cf. Isaak et al., 2017, 2020). This map can presently be used to predict and manage cold-water stream habitats, with potential for regular multi-annual updates.

To enable unbiased comparison among stations from time series with gaps, we used a climate correction of $T_w$ based on regression with $T_{air}$. The efficacy of regressions between $T_w$ and $T_{air}$ is well understood (Ducharne, 2008; Segura et al., 2015, Moatar and Gailhard, 2006), and the approach can be easily transferred to other stations and regions. On the other hand, a cautious approach is required for such regressions, because they assume seasonal correlation between $T_{air}$ and $T_w$, and can therefore only apply to rivers having natural seasonal dynamics, without dam release, thermal peaking, or major weirs regulating the flow of streams (Bruno et al., 2013; Chandesris et al., 2019; Seyedhashemi et al., 2021; Casado et al., 2013). Moreover, our rolling window approach to $T_w$–$T_{air}$ regression suggests that the watershed size may be an important factor in developing these relationships, with regressions for larger rivers benefitting from longer lags in $T_{air}$ (Fig. B2). However, where possible, climate correction makes it possible to significantly reduce the biases in $T_p$ estimates when compared to those made using observed data without climate corrections (Figs. 2 and 3). The biases are reduced when the number of years of observation available is less than four. Beyond this limit, the meteorological variability specific to each year is sufficient to estimate $T_p$, which confirms results obtained by (Jones and Schmidt, 2018). Moreover, climate correction reduces overestimation of $T_p$, reinforcing the importance of taking into account these climatic corrections of temperature metrics even if it only slightly affects the majority of rivers (Isaak et al., 2017)

**4.2 Spatial extrapolation of $T_p$ is consistent and best predicted with a random forest model**

All four statistical models achieved similar predictive performance, but the RF model exhibited marginal improvements over the others (Fig. 3). The MM approach only slightly improves performance in cross-validation in comparison with RF (vis-à-vis the NSE criterion), so it is unlikely to be a useful approach in future applications due to its complexity. Moreover, whereas the multi-model has the best performance, it lacks the explanatory power and relative simplicity of the other approaches. In contrast, a potential benefit of the multi-model approach is that by leveraging multiple approaches, it can compensate for errors particular to individual models.

Overall, our model performances (NSE = 0.75, RMSE <2) were of the same order as other large scale stream temperature studies found in the literature (cf. Segura et al., 2015; Daigle et al., 2010; Wehrly et al., 2009), suggesting that our approach is reasonable and broadly applicable. Indeed, literature ranges of RMSE for monthly stream thermal maxima are consistent with values we obtained here (e.g., 0.9–2.1°C for 16 sites across Canada (Daigle et al. 2010) and 2.0–2.3°C for 1131 sites across USA (Wehrly et al., 2009)), suggesting a general accuracy limit to current modeling approaches. This is likely because RF models can be prone to overfitting such that they can accurately predict a set of observations, but their performance may decline when predictions are made at

unsampled locations. They also have less robust means of model selection and significance testing than the much

simpler multiple linear regression approach.

In spatial extrapolation, the $T_p$ estimates are globally consistent between the models and the same spatial structures are found regardless of the approach used (Figure 6). We observed divergences between the models in particular for $T_p$ less than 14°C where ANN tends to predict coldest $T_p$ for more reaches compared to other models (Fig. B4). Still, because our analysis is limited to 2009–2017, these model differences may diminish as $T_w$ measurements

grow in time and space. We further note that the performance of the modeling techniques used here was less than that of spatial stream-network models (SSNs) applied to similar temperature datasets, which typically have $R^2 \sim 0.90$ and RMSE~1.0°C (Isaak et al. 2017). We tested a SSN model on a small region well covered by data (9000 km², 92 stations) for a robust estimation of parameters with the R package *SSN* (see Fig. B5 and Table B1), and indeed; SSN had reduced bias relative to than random forest model (absolute bias decreased by 0.2°C). By

comparing the observed and estimated values, we can see that compared to the SSN, the RF model tends to underestimate the high values (on major rivers) and to overestimate the low values (on smaller reaches). Still, the spatial patterns are very consistent among the two approaches, though there are important differences between the SSN and RF model estimates which can be +/- 2°C. However, SSNs are labor intensive to apply in comparison to non-geospatial techniques and require specialized geospatial algorithms for fitting. Hence, despite lower accuracy,

the approaches used here may be useful in more generic use cases when geospatial data and computing time are limiting.

**4.3 Drivers of thermal peak depend on model structure**

We observed clear divergence of variable importance for the estimation of $T_p$ among the ANN, RF, and REG models. The two most relevant variables in RF and REG were watershed area and mean summer air temperature,

consistent with other studies (Laanaya et al., 2017; McGarvey et al., 2018; Moore et al, 2010, 2013). Drainage area also emerged as one of the most important variables driving thermal peaks, behind watershed elevation and slope, in a recent regional study (Johnson et al, 2020). Likewise, distance from source and Strahler order, which directly correlate with drainage area, are known to be important drivers of thermal peaks (Hrachowitz et al, 2010; Imholt et al, 2013; Ducharne, 2008, Mohseni et Stephan, 1999). The clear rationale for this common effect is that

longer residence times in large rivers allow more time for temperature equilibration with the atmosphere compared to small rivers (Beaufort et al., 2020a; Mohseni et al., 1998, Fullerton et al., 2015). Large rivers are also less shaded by riparian vegetation and are less influenced by groundwater inflows, which compounds residence time effects. Importantly, RF had a much more even distribution of variable importances relative to the other models structures, which is likely due to its non-linear structure. In contrast, for ANN, the most important variables were minimum

monthly specific discharge (43%) followed by riparian vegetation cover (23%). Higher minimum flows imply a consistent groundwater supply, leading to cool surface waters in summer (Hannah et al., 2004; Kelleher et al., 2012; Lalot et al., 2015). Similarly, greater riparian vegetation implies more shading and reduced temperature increases from solar radiation (Dugdale et al., 2018; Loicq et al., 2018; Moore et al., 2013). These differences in relevance between the variables for each model underlines the importance of using several approaches and shows

that the multi-model approach makes it possible to take into account all these across-model divergences. We caution however, that while predictability may increase, interpretation of variable importance in multi-models is

complex, and their utility may therefore decrease when project goals are focused on critical variables for stream temperature habitat restoration.

**4.4 Air temperature is not an appropriate proxy for stream temperature**

Estimates of $T_p$ produced by stream temperature were clearly more accurate than those produced by air temperature (Figs. 3 and 5). In cross-validation, $T_{p,air}$ overestimated observed $T_p$ by more than 2°C, could not differentiate stream temperature among regions (Figs. 6 and 7), and could not differentiate large rivers from small rivers. These results highlight the need to account for stream-specific temperature estimates, especially when such $T_{p,air}$ overestimates inflate potential ecological change (cf. Kirk and Rahel 2022). Our results further demonstrated that

this overestimation of $T_{p,air}$ was greatest in smaller rivers, and that these biases were amplified in streams with significant groundwater contribution (Fig. 7), implying that groundwater buffers effects of increasing $T_{air}$. This aligns with recent results that found sites with deep groundwater contributions are much less likely to exhibit increasing summer stream temperatures compared to sites with shallow groundwater contributions (Hare et al. 2021). However, we note that the influence of $q_{min}$ on $T_{p,air}$–$T_{p,w}$ is weak up to values of approximately 5 L s$^{-1}$ km$^{-}$

$^2$ (Fig. 7b), which is in accordance with recent work in the same region (Beaufort et al., 2019). This small effect may in part be because $q_{min}$ is not an effective proxy for groundwater contributions; the base flow index is likely more appropriate (Kelleher et al, 2012; O'Driscoll & DeWalle, 2004, Hare et al, 2021; Johnson et al, 2020), but we were unable to obtain this parameter in this work at a national scale.

Overall, this work clearly demonstrates that $T_{air}$ is an inappropriate proxy for $T_w$, with important implications for

ecological studies, especially those that consider temperature tolerance thresholds of aquatic species. Species distribution models may need to use $T_{air}$ instead of $T_w$ because data from $T_w$ are not sufficient in the regions studied (McGarvey et al., 2018). It is therefore important to introduce $T_w$ in input of these models rather than $T_{air}$ in order to limit the biases linked to the poor spatial representation of $T_{air}$.

This research further emphasizes the importance of spatial scale and heterogeneity for water temperature studies.

As streams increase in size, they become more coupled to air temperature dynamics. Hence, smaller reaches more influenced by groundwater inputs and vegetation may serve as "climate refugia" for ectotherms species especially in the context of climate change. Therefore, choosing the best predictors at a spatial scale for thermal peaks and other thermal regime metrics is essential to accurately predict and manage future stream cold-water habitat.

**4. Conclusion**

Stream thermal maxima are essential controls on aquatic ecosystems, but our understanding of their spatial distribution and thus our ability to adequately manage them accordingly is limited by data availability and simple metrics applicable at large scales. To address these gaps, we created a publicly available, harmonized dataset of stream temperature that can used by ecologists in France and scientists more broadly. We then developed a simple, ecologically relevant metric–the thermal peak, $T_p$– that can be extrapolated at large scales, even when data are

sparse. We developed an innovative climate correction method to reduce biases related to such data sparseness, when applied to summer data. The $T_p$ provides an important perspective on the magnitude of thermal maximums during summer, but development of additional metrics such as threshold exceedance frequency, duration, and timing will continue to grow our understanding of stream temperature behavior under climate change. However, development of these metrics will require longer and more spatially explicit time series. Hence, we argue that to

improve our capacity to manage and benefit from aquatic ecosystems, it is critical to continue and expand our stream temperature measurement networks.

## 5. Appendix A: Classification of river flow regime for France

Sauquet et al. (2008) defined a classification system for river discharge based on the mean monthly runoff pattern (Fig. A1). Using this classification system, they published a map with each river reach assigned to one class along

the main river network.

Generally, the uppermost basins located in mountainous areas display glacial and snowmelt-dominated regimes (Group 1–3). The lower the outlet, the lower the contributions of snowmelt to runoff. Group 4 to 6 were defined to be in the "transition" regime, where the seasonal variation in streamflow is affected as much by precipitation timing as by air temperature and topographic influences on snowpack formation and snowmelt timing. In this

regime, high flows are typically observed in spring. Group 7 to 12 are rainfall-dominated regimes. These six rainfall-dominated groups differ along an axis of maximum and minimum monthly discharges. For instance, group 7 is characterized by very low flow in summer, reflecting the lack of deep groundwater storages in the catchment. In contrast, Group 12 exhibits nearly uniform flows through most of the year, and these river reaches are typically found where large aquifers moderate flows.

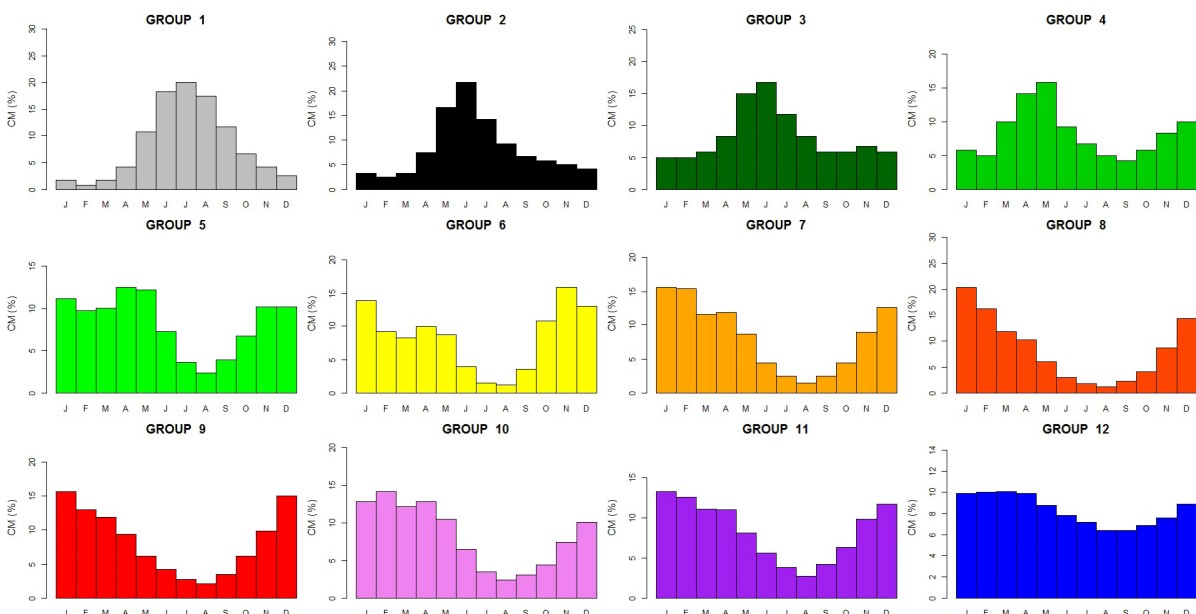


**Figure A1.** Reference dimensionless hydrographs representative of the classification of river flow regime for France (after Sauquet et al., 2008). The y-axis (CM) represent the percent of each month contributing to total annual runoff, equal to monthly runoff (mm) divided by the mean annual runoff (mm).

## 6. Appendix B

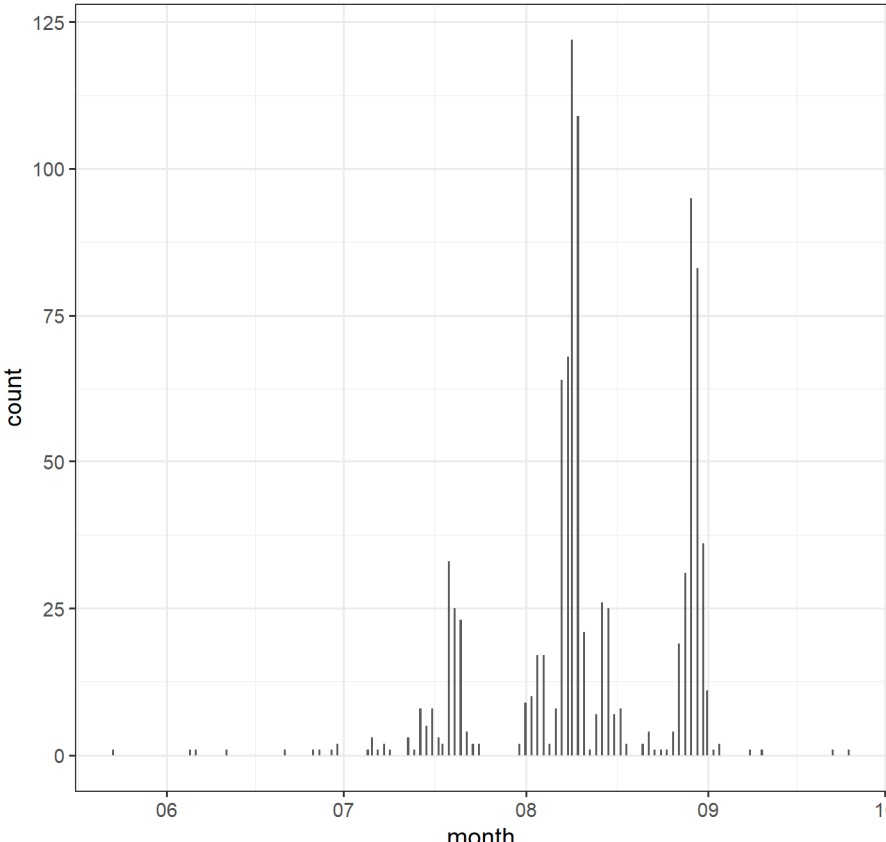


**Figure B1.**      Histogram of the median date for the hottest 30-day periods across all sites that had annual data. Thirty day rolling windows of mean daily stream temperature were calculated across the entire time series and for each site, and the 30-day windows with the maximum values were selected here. The median date here refers to day 15 in the 30 day period.

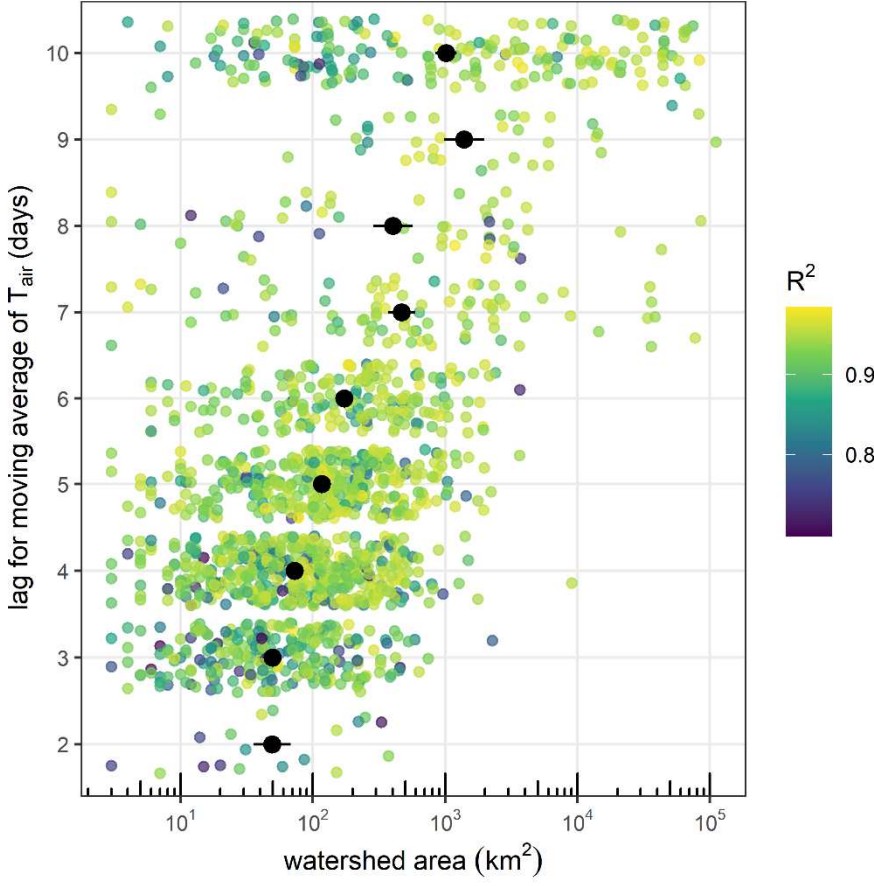


**Figure B2.**        Daily lags for $T_{air}$ that produce the highest $R^2$ (colors) for a regression between $T_w$ and a right-aligned moving average of $T_{air}$ at each site as a function of watershed area. The moving average lag whose regression produced the highest $R^2$ was used to fill gaps in the time series of $T_w$ for each station. Each point represents a station, with the x-axis being the watershed area at that station, the y-axis being the lag in $T_{air}$ that

produced the best regression. Points are jittered for visual clarity; only integer lag values were possible. Black points indicate the mean and standard error (horizontal bars) of watershed areas within each possible lag (2–10 days).

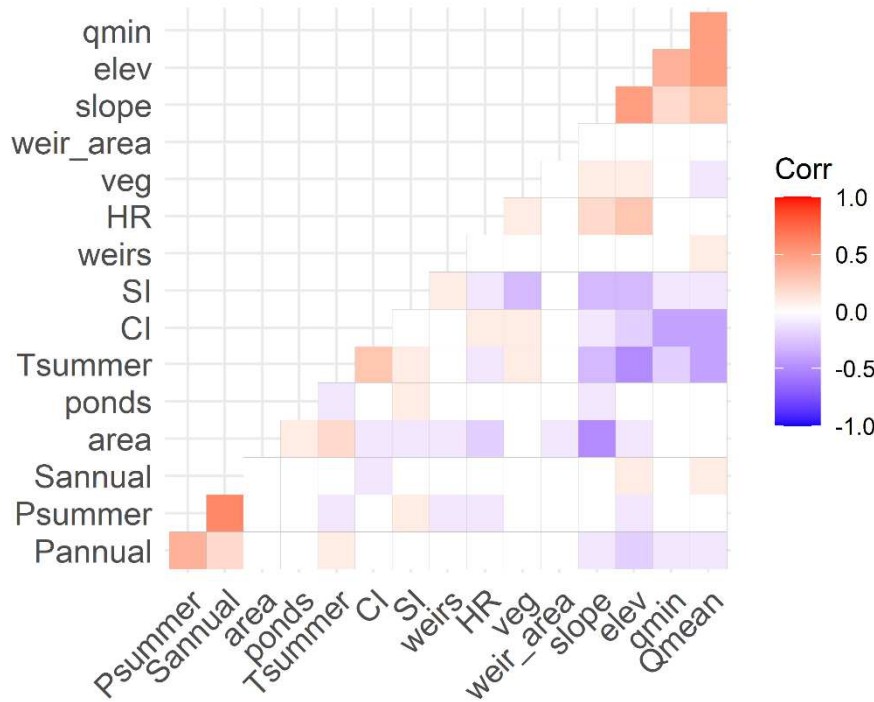

**Figure B3.** Correlation matrix of all 16 considered environmental variables prior to selection in the thermal peak analysis. All variables had collinearity values less than 0.6 or greater than -0.6.

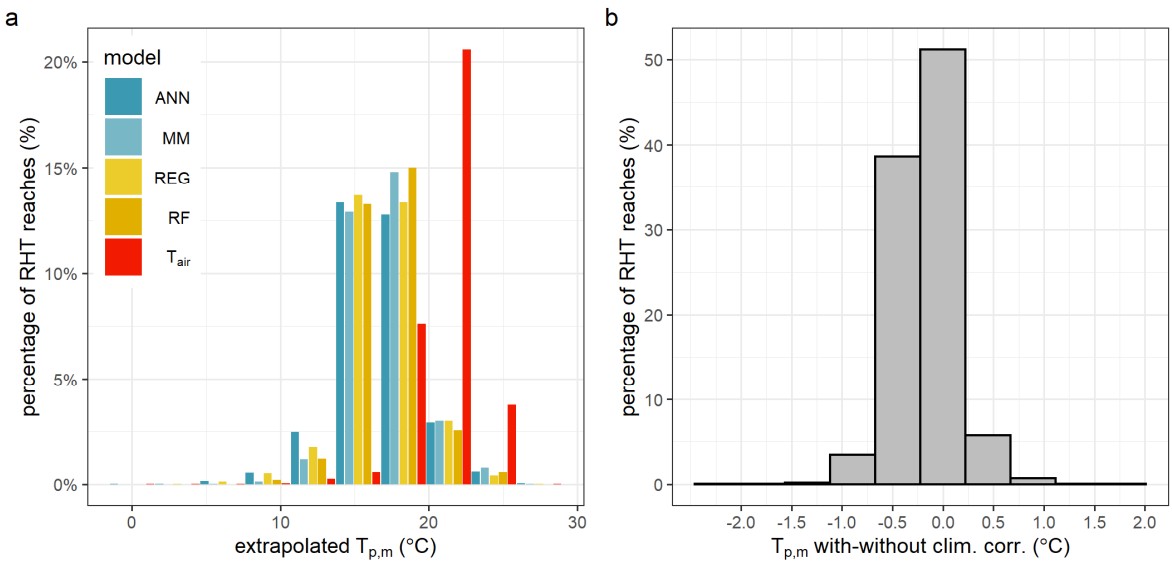

**Figure B4.** Distributions of $T_{p,m}$ extrapolated to all RHT network reaches. a) Histogram of models (10 bins equally spaced between 0°C and 30°C) according their extrapolated $T_p$ value, colored by model, and b) differences between $T_{p,m}$ calculated with and without climate corrections on the 1,630 stations with gaps in their data.

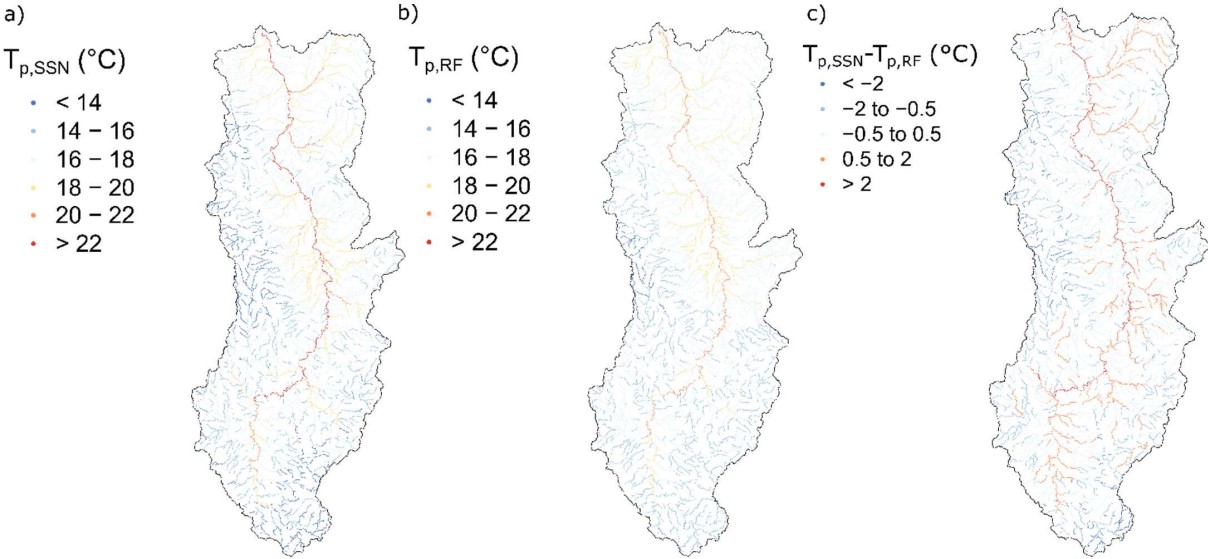


**Figure B5.**     Figure comparing modeled thermal peak ($T_{p,m}$) estimates from (a) SSN, (b) RF, and (c) the difference between RF and SSN. The SSN model here is from a benchmark test on a small region well covered by data (9000 km², 92 stations) for a robust estimation of parameters with the R package *SSN*. SSN had reduced bias relative to than random forest model (absolute bias decreased by 0.2°C). Additionally, by comparing the observed and estimated values, we can see that RF tends to underestimate the high values and to overestimate the low values. Unfortunately, due to the lack of an RHT with upstream-downstream information, we could not apply at the scale of the whole watershed. Still, the spatial patterns are very consistent among the two approaches, though there are important differences between the SSN and RF model estimates which can be +/-2°C. The estimates of the SSN model are generally warmer than those of RF on the main major river axes and colder on the small tributaries. This is consistent also with observations. So, while the presented models may not be optimal, we are confident the spatial patterns are correct.

**Table B1.** Comparison of model performance metrics for the region tested in Figure B5

| Model | RMSE (-) | NSE (-) | Bias (°C) | Absolute bias (°C) |
|---|---|---|---|---|
| RF | 1.24 | 0.44 | 0.01 | 0.95 |
| SSN | 0.99 | 0.81 | 0.05 | 0.76 |

**Code/Data Availability**

Water temperature data provided by OFB can be downloaded from http://www.naiades.eaufrance.fr/acces-donnees#/temperature.

Data generated by this work can be found at: https://thermie-rivieres.inrae.fr. Additional data and code are from the corresponding author upon reasonable request.


**Author Contributions**

The paper has been authored by AB with contributions from all the co-authors. AB, FM, JD and ES contributed to the conceptualization and methodology. AB and JD performed the formal analysis. ES contributed hydrology metrics. AB, JD, ES, and FM contributed to the writing and revision of the manuscript.


**Competing Interests**

The contact author has declared that neither they nor their co-authors have any competing interests.

**Acknowledgements**

The authors thank Jan Seibert, the editor of Hydrology and Earth System Sciences, and the three reviewers for their valuable comments, suggestions, and positive feedback on the manuscript. The authors also thank the OFB, the many regional fishing organizations, DREAL, and EDF for the temperature data. The SAFRAN database was provided by the French national meteorological service (Météo-France).

**Financial support**

The research project was funded by the French Agency for Biodiversity (OFB).

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
