# Peer review of "Spatial extrapolation of stream thermal peaks using heterogeneous time series at a national scale"

_Hydrology and Earth System Sciences, 2021_

## Referee Comment (RC2)

In Beaufort *et al.* the authors compiled stream temperature data for the entirety of France and Corsica, calculated an ecologically relevant summarizing metric ("the thermal peak"), and compared predictive models and an air proxy to determine the best model structure and predictor variables for extrapolating the thermal peak metric to all rivers in France/Corsica. Although the work done within is a valuable contribution, the manuscript would be improved by (1) altering the climate-change framing of the paper, (2) including a specific question or hypothesis, and (3) increasing specificity and clarity of the presentation. Regarding the first point, the paper frames the research as important for understanding stream thermal regimes under climate change. However, the authors calculate a metric which does not help us understand how climate change alters stream temperatures. This is a major issue. Additionally, there lacks a clear question or hypothesis in the introduction – another major issue.

(1) Comments regarding emphasis on climate change:

The introduction leads me to think that the authors would be explaining how stream temperature is changing in France/Corsica as a result of climate change. For example, see:

Line 36-37: "However, the magnitude and direction of these expected changes will depend strongly on patterns of stream temperature change"

Line 38: "Hence it is critical to describe and analyze the spatiotemporal variability of river thermal regimes."

Line 71-74: "Indeed, stream temperature metrics that focus on extreme periods […] are likely adequate to understand trends of increasing pressures on aquatic ecosystems."

And in the Methods:

Line 102-103: "To address ecologically meaningful temperature metrics under climate change, we focused on the two hottest stream temperature months, …"

However, the metric the authors calculate does not describe temporal stream temperature changes. The described calculation of the thermal peak is an average of peak temperature across years for each logging station. The metric is thus only useful for understanding the spatial pattern of the thermal peak across France/Corsica for the years in this study. Although the compiled database itself may be a useful source for studies on climate change, the thermal peak metric calculated in this study does not describe changes in stream temperature through time. Indeed, the authors even admit to this shortcoming in the discussion:

Line 315-316: "The downside of the current approach is that it remains based on interannual metrics. Indeed, the non-concomitance of the time series does not allow us to compare extreme years (hot vs. cold)."

More emphasis should be put on the thermal peak being a good comparison metric for rivers across space and may help managers understand which rivers currently exceed thermal tolerances of important biota.

(2) Comments regarding lack of question/hypothesis:
    a. An objective is stated in the introduction (Line 73-74) but no specific question, hypothesis or prediction is explained. This left me wondering throughout the paper what the purpose of the research was. What questions motivated this research? What did you expect to learn from research at hand? What do you expect the thermal peak tell us about rivers in France/Corsica?
    b. Section 4.3 of the discussion aims to attach meaning to the most important explanatory variables across each model. There was no hypothesis about which explanatory variables were thought to be the most important predictors, therefore it seems the authors are fishing for an explanation without having a clear question, hypothesis or prediction about these explanatory variables. Since the author's purpose for making these models was to predict the thermal peak across all rivers of France/Corsica, it is inappropriate to interpret the explanatory variables in this way.

(3) Comments regarding improvements on clarity and specificity:
    a. Line 88, sentence starting with "Hence, our main challenge was to pool...": This sentence was difficult to understand. Improve with a more thorough explanation, possibly breaking up this sentence into multiple.
    b. Line 99, "data were … averaged into mean daily stream temperature data". Do you mean "daily mean" here? It is unclear if it is meant to be the mean daily temperature (averaged across years for a particular day) or the daily mean temperature (average temperature of a specific date).
    c. Line 100, "1) hourly $T_w$ anomalies". If $T_w$ is defined as the daily mean temperature, how can you see *hourly* anomalies if there is one mean temperature per day? May need to define another variable to use here.
    d. Line 101, "$T_{air}$": Need to define this variable because it is the first time it is used.
    e. Section 2.2, first paragraph. Why hold back on defining the "simple metric" within the first sentence? It is odd to hold back on defining the metric and then reveal the big mystery of what the metric is.
    f. Section 2.2, second paragraph. $\overline{T_{w,30}}$ is defined as the "mean temperature of the 30 hottest consecutive days of each year". However, the data have already be constrained to July and August. How can you be sure that the 30 hottest consecutive days occur completely within the window between July 1 and August 31? In the last sentence you conclude that the hottest day of the year always occurred between July 28 and August 30. This implies that for the temperature time-series in which August 30[th] is the hottest day of the year, the 30 hottest consecutive days would include days in September. So, $\overline{T_{w,30}}$ should be re-defined in the paper as the mean temperature of the 30 hottest consecutive days in July and August. Be more specific about defining $\overline{T_{w,30}}$.
    g. Line 164, "SAFRAN data". Define what this data is.
    h. Line 198, "… all possible variables." Very broad. More specificity would be helpful, such as "all possible variables characterized in Table 2."
    i. Line 236-238, "This bias … in the regressions." These sentences are difficult to understand. Be more specific to improve clarity. The ending of one sentence is "…with

only one year of observation.", and the next sentence begins, "In contrast, when there is only one year available…". These aren't contrasting clauses, thus making it difficult to understand what is being contrasted.

    j.   Sentence starting on line 260, ending line 263, "The two most relevant…". Break into 2 sentences to improve clarity.

    k.   Line 281: use of 14°C. Explain why 14°C important and why this value chosen as a cut-off point.

    l.   Figure 6C. Figure 6c shows probability distributions for extrapolated thermal peak. Firstly, the bins are not of equal size: the first bin (labeled <14) includes all temperatures between 6.3 (minimum temperature reported on line 277) and 14, a range of 7.7 degrees; the second through fifth bin have ranges of 3 degrees; and the final bin includes temperatures from 22 to 27, a range of 5 degrees. The 14- and 22-degree cut-off is not explained and thus seems arbitrary. Secondly, since temperature is continuous, it shouldn't be binned in this way, unless for some specific reason. Continuous distributions should be portrayed as "probability density functions". Each model would then have a continuous distribution, and each can be plotted on the same graph for comparison.

    m.   Line 301, first sentence in discussion. Be more specific about the timeframe of the dataset: "the largest, regional *summertime* stream temperature datasets"

    n.   Line 326. Which "metrics based on observations", specifically, are you comparing to estimates of $T_p$?

    o.   Line 328. Further explain what is meant by "when applied at scale". How does that relate to figure 6B?

    p.   Line 352: "Higher minimum flows" would make more sense than "Larger minimum flows"

    q.   Line 354: Change "greater" to "more". "Great" implies a quality of shade and not a quantity. "More" implies a quantity.

(4)  Technical Corrections

    a.   Line 21: Use of "However" does not belong because this sentence doesn't relate to previous

    b.   Line 27: "…,  decomposition rates,  dictates animal … "

    c.   Line 33: "and levels" is repetitious with flows

    d.   Line 96: "… were previously…" change to "have been". "were previously" implies that a different study excluded these data. If that is the case, cite that study.

    e.   Line 155: "is not clear" should be "was not clear"

    f.   Line 221, equation 7: regression coefficient "a" is in the equation twice and "c" is missing.

    g.   Line 229: "cross-validation  100 times"

    h.   Line 321: insert "between" between "correlation" and "$T_{air}$"

    i.   Line 325: remove "particularly"

    j.   Line 326: remove "a robust interannual metric". Only need to say "… sufficient to estimate $T_p$"

k.  Line 329: change "to" to "of" in "climate corrections to temperature metrics"
l.  Line 352: change "following" to "followed"
m.  Line 373: change "representativeness" to "representation"

---

## Author Comment (AC1)

Journal: HESS
Title: The thermal peak: A simple stream temperature metric at regional scale
Author(s): Aurélien Beaufort et al.
MS No.: hess-2021-218
MS Type: Research article

**Responses to Referees**

**Anonymous Referee #1:**

We thank Referee #1 for their detailed review of our manuscript. We have broken out your individual comments, which are numbered, and responded to each accordingly in blue. We hope that our comments address and clarify any issues or concerns that they may have.

**Comments**

Beaufort et al. address an important topic in their manuscript, "The thermal peak: A simple stream temperature metrics at the regional scale"—namely, how does one develop accurate stream temperature information for an area the size of France that could be used in climate assessments or research on thermal ecology of lotic species? To accomplish this task, the author assemble a large national database of temperature measurements, summarize these records using a single metric called "The thermal peak', link this metric to site and watershed level descriptions derived from GIS sources, and then model the dataset using four different approaches to compare and contrast the outcomes. What the authors have undertaken is ambitious and to be commended, but I do have several reservations about the manuscript in its present form, as outlined below, that could be addressed to improve its overall quality.

1. Consider a revision of the title so that it better represents the research question and issues at hand because it currently is focused a relatively minor methodological issue relating to how temperature records are summarized.

   *We agree with the reviewer and are considering alternative titles including the following:*

   *Spatial reconstruction of simple stream temperature metric at regional scale: the thermal peak*

   *Spatial extrapolation of stream temperature thermal peaks using heterogeneous time series*

2. Abstract. Add or revise the lead sentence to that it also frames the research question more broadly. For example, why do we care about or need stream temperature information?

Climate change, water quality standards, thermal ecology could all be drawn on as motivating factors. I also disagree with the claim made in the lead sentence, that "spatiotemporally comprehensive stream temperature datasets are rare…" because the literature is full of stream temperature studies, and there are now many grassroots and state sanctioned monitoring programs. What's really the issue is that the data are scattered among many entities and rarely organized into a central database. The fact that the authors have built such a large database for France during the course of this research shows that stream temperature data are common, and the database itself is a valuable contribution.

We agree and will change the text as recommended. For example: "Spatial reconstruction of stream temperature is relevant to water quality and fisheries management, yet large, regional scale datasets are rare because data are decentralized and not coherent scattered among many entities."

3. Introduction, line 40. There is mention made here of thermal regimes and their components (frequency, magnitude, etc) and that continuous records, preferably of extended length are needed for accurate regime description. I disagree that this is the case as lengthy records are primarily useful for trend detection, as might be the case when describing the effects of climate change. More importantly, from the perspective of this manuscript is that many of the dozens of metrics that are often used to describe thermal regimes are strongly correlated. Thus, it is valid to focus on one (or a small set) summary metric, model it, and know that your representing a lot of the information about overall thermal regimes. This is the point you should make here, these three papers all provide good examples of the strong correlations among thermal metrics. Steel et al. 2016. Spatial and temporal variation of water temperature regimes on the Snoqualmie River network. JAWRA Journal of the American Water Resources Association, 52:769-787; Rivers-Moore et al. 2013. Towards Setting Environmental Water Temperature Guidelines: A South African Example. Journal of Environmental Management 128: 380–92; Isaak et al. 2020. Thermal regimes of perennial rivers and streams in the Western United States. Journal of the American Water Resources Association, 56:842-867.

We thank the reviewer for this important clarification and will rephrase the text as recommended and include the suggested literature. We will replace "However, these

metrics can be accurately determined only if continuous time series of stream temperature are available (Jones and Schmidt, 2018)." with

"However, many of these stream temperature metrics are strongly correlated (references), implying the utility of a single metric to understand stream temperature regimes."

4. Methods line 90. The authors state that "the large spatial and temporal heterogeneity of the monitoring data precluded application of spatial autocorrelation models…" This isn't an accurate statement, SSN models are perfectly suited to this type of temperature database, as two of the studies cited by the authors demonstrate (Detenbeck et al. 2017 and Isaak et al. 2017). Nonetheless, it's fine not to use SSN models and rely on other approaches so I would just delete this sentence.

We agree and will remove the referenced text: "The large spatial and temporal heterogeneity of the monitoring data precluded application of spatial autocorrelation methods, and we have therefore chosen to consider only non-spatial statistical models."

5. Line 118, Thermal peak metric. Derivation of this metric seems far more complicated than it needed to be, while also discarding valuable information about inter-annual variability by averaging over multiple years of observations at individual sites. Because the dates of the 30 warmest days will be different each year, it also adds some inconsistencies and creates complexities for processing the temperature records. The same information about the thermal regimes could have been obtained from a simple mean July or mean August temperature metric.

We agree with the Reviewer that the thermal peak calculation is somewhat involved, but we point out here that we did not know a priori that July and August would be the hottest months. The method used here is analogous to the one used for developing macroinvertebrate/fish typologies of France (Buisson and Grenouillet 2009), so we will include this rationale and relevant citations in the revised methods. To support our approach, we will also include a sensitivity analysis on the sites with annual data where we compare the $T_p$ of the annual time series with the $T_p$ limited to July and August. Finally, we will add text in Discussion to suggest simpler approaches.

6. Methods, line 170. Reference is made here to a principal components analysis but it's unclear how this was employed or the effects it had on excluding variables from consideration.

We removed this reference to the PCA as it was unnecessary. Instead, we will include the following correlation matrix in Supplementary Material to illustrate the independence of variables used in this analysis. Please also see our related response to comment 7.

[Figure]

7. Methods, Table 2, explanatory variables. It's not clear from the manuscript text how some of these variables would affect stream temperature. Please expand this table with another field labeled "Hypothesized effect" and briefly explain the rationale for considering each variable in the models, preferably with supporting citations.

We agree that this would greatly improve the clarity of the variable selection. Please see below for a revised table.

**Table 2.** List of explanatory variables used in models.

| Category | Variable | Notation | Source | Possible effect |
|---|---|---|---|---|
| Climate | Mean annual precipitation (2009 –2017) [mm] | $P_{annual}$ | SAFRAN | Contrast between climatic regimes (Moore et al, 2013) |
| | Mean summer precipitation, July–August (2009 –2017) [mm] | $P_{summer}$ | SAFRAN | Influence of heat budget Caissie, 2006 |
| | Mean annual snow accumulation (2009 –2017) [mm] | $S_{annual}$ | | Heat budget, meltwater influence, Caissie, 2006, Webb et al, 2008 |
| | Mean summer air temperature, July–August (2009 –2017) [°C] | $T_{summer}$ | | Positive effect related to heat budget, relative Moore et al, 2013 (index of the thermal summer climate) |
| | | | SAFRAN | |
| | | | SAFRAN | |
| Hydrology | Mean annual specific discharges [L s$^{-1}$ km$^{-2}$] | $Q_{mean}$ | | Thermal capacity influence, Caissie, 2006 |
| | Mean monthly minimum specific discharge with a return period of 5 years* [L s$^{-1}$ km$^{-2}$] | $q_{min}$ | RHT | Proxy of base flow index, Chang and Psaris, 2013 |
| | Concavity index† [-] | CI | RHT | Proxy of water storage in the catchment (snow or groundwater), tested in this paper |
| | Hydrological regime‡ [-] | HR | RHT | Contrast between hydrological regimes, tested in this paper |
| | | | RHT | |
| | | | RHT | |
| Catchment characteristics | Mean catchment elevation [m] | elev | RHT | Negative effect given the relation with air temperature (Isaak and Hubert, 2001) |
| | Drainage area [km²] | area | RHT | Proxy for width-depth ratio of streams (Hrachowitz et al, 2010), and thermal capacity (Imholt et al, 2013) |
| | Mean streams slope over the catchment [m km$^{-1}$] | slope | RHT | Affect river hydraulics and thus thermal advection and exposure time to incoming radiation (Daigle et al, 2010) |
| | Riparian vegetation cover ratio in 10 meters buffer (%)** | veg | SYRAH | Negative effect, as decrease exposure to diurnal radiation (Moore et al, 2005) |
| | Linear weir density upstream of stations (# km$^{-1}$)** | | | Potentially heating effect (Chandesris et al, 2019) |
| | Areal weir density upstream of stations (# km$^{-2}$)** | weirs | SYRAH | Potentially heating effect (Chandesris et al, 2019) |
| | Pond cover ratio upstream of stations (%)** | weir area ponds | SYRAH | Potentially heating effect when ponds and shallow reservoirs release warm water from overflow (Seyedhasemi et al, 2021) |
| | Stream incision class ** | SI | SYRAH | Potential proxy of exchange of water with hyporheic environment, Webb et al, 2008 |
| | | | SYRAH | |

**8.** Methods, lines 197-215 describing the analysis techniques. Please provide more detail. The minimum requirement is providing enough information that a knowledgeable reader could replicate the analysis. In the case of the multiple regression, for example, how was model selection performed (e.g., AIC based, stepwise, best subsets, etc.). Was the potential for problematic multicollinearity assessed by removing highly correlated explanatory variables?

We will improve the clarity and detail on the techniques used in this section. For the multiple regression, we did not use any variable selection techniques. Our goal was not to have the best possible regression, but instead to use the already determined independent

variables (see response to comment 6) to compare across modelling techniques (i.e., regression, ANN, random forest, multi-model). We will add text to make this more clear here and at the end of the Introduction.

9. Methods lines 218-222 describing the multi-model combination. It's unclear how this was done exactly. The authors state, "estimates from each previously described model were…" Usually parameters are estimated, so I think you really mean "temperature predictions from each previously described model were…" Moreover, those prediction combinations were presumably done for each reach within the network, so there should be an "i" subscript in the equation notation to denote this.

Indeed, we intended "predictions" instead of "parameters", which we will clarify. The reviewer is also correct with reference to the predictions being reach-specific; we will therefore include an "i" index in the equation. To clarify with regard to the mutli-model approach. the predictions made by the three models are included and each model prediction is weighted with a coefficient to match the observations as closely as possible. Hence, the coefficients are calculated only in relation to observations, so only where there are stations. Then, using this equation (7) with calculated coefficients, we extrapolate to simulate reach-$T_p$ at the network scale (see Figure 5). We will be more clear on this in the Methods section.

Also useful for comparing the models would be a multipanel figure containing a series of bivariate scatterplots showing the pairwise predictions from each combination of the models with the associated correlations shown. These correlations are quite high presumably, but one could also further explore the discrepancies between model predictions by analyzing the residual differences relative to the predictor variables.

We agree that this analysis would be useful and will include a version of it in the Supplementary Material. We already have some prototypes of this analysis that are spatially explicit, which may be more informative than the suggested scatterplots, which we include below.

[Figure]

*(a) Multiple regression model vs. observations ; (b) random forest model vs. observations ; (c) ANN model versus observations and (d) Multi-model combination versus observations*

**10.** Results lines 244-245. It's unclear where the air temperature model predictions of stream temperature came from. Is the air temperature model a simple linear regression with air temperature the single predictor of stream temperature? If so, it should be mentioned and described in the preceding methods section with the other model types.

We agree that this was not clear and will correct this in Methods text and the variable definition table. To clarify here, there is no regression. The air temperature predictions are simply SAFRAN reanalysis data.

**11.** Results, lines 259-265. Relevance of explanatory variables in the models. Inconsistent terminology in this section makes it difficult to understand how the explanatory variables are being assessed. Initial reference is given to "Explanatory power", later in the paragraph "cumulative importance" is referenced, and the accompanying Figure 4 refers to "relative importance." Are these all the same things and/or do they reference the r2 statistic? Please clarify. Also, it would be useful to expand Figure 4 to see the effects of all the variables that were important contributors to each model, and to know what the total explanatory power was of each model.

We will simplify this terminology and clarify in the Methods to better present this information. Indeed, throughout this section we are referring to the same variable importance as described in part 2.4.2–2.4.4. These importance values are then summed to get cumulative importance; it was therefore necessary to standardize these importance terms. We do not check the explanatory power of the variables in the prediction itself, but we look at which variable each model used to obtain its prediction. We have chosen not to present the many other variables in this figure for both visual clarity, and because the other variables' importance are negligible, as is evident by the high cumulative importance of the variables shown. We will, however, expand on the total explanatory power of each model and discuss minor relevance of the other variables in the Discussion.

**12.** Discussion section, lines 315-316. Because of the way the thermal peak metric was calculated and model fits were conducted, by using temperature observations averaged across years, the ability to estimate inter-annual effects due to variability in air temperatures and discharge was lost. However, the stream temperature dataset certainly contains that information and it may be important to recognize and estimate in future model iterations because it can enable climate change forecasting. A technique for retaining both spatial and interannual temporal variation in model fits to similar stream temperature datasets was employed in both the Isaak et al. publications the authors cite and might be referenced in this section of the discussion.

We agree and will reference this shortcoming in the Discussion while citing these relevant publications.

**13.** Discussion section lines 330-341 concerning spatial extrapolation by random forest models. It would be useful to expand this section and bring more balance to it with a discussion of the pros/cons of the various model types. For example, random forest models are easy to apply but are also generally known to overfit such that they can accurately predict a set of observations but may see performance declines when predictions are made at unsampled locations. They also have less robust means of model selection and significance testing than say multiple linear regressions. In all cases, the performance of the modeling techniques used here was less than that of SSNs applied to similar temperature datasets, which typically have r2 ~ 0.90 and RMSE ~ 1.0 C but SSNs are labor intensive to apply in comparison to non-geospatial techniques and require specialized geospatial skills to fit.

We agree that SSNs are useful in these applications, and indeed, we conducted some benchmark tests on small region well covered by data (9000 km², 92 stations) for a robust estimation of parameters with the R package *SSN* (see figures below). SSN performed better than the other methods (decreased by 0.2°C for SSN model compared to random forest), which was encouraging. Additionally, by comparing the observed and estimated values, we can see that RF tends to underestimate the high values and to overestimate the low values. Still, the spatial patterns are very consistent among the two approaches, though there are important differences between the SSN and RF model estimates which can be +/- 2°C. The estimates of the SSN model are generally warmer than those of RF on the main major river axes and colder on the small tributaries. This is consistent also with observations. Unfortunately, due to the lack of an RHT with upstream-downstream information, we could not apply SSN at the scale of France.

So, while the presented models may not be optimal, we are confident the spatial patterns are correct. We will include a more detailed discussion of the pros/cons of the different models with the possibility of SSNs.

[Figure]

| Model | RMSE | NSE | Biais | BiaisAbs |
|-------|------|-----|-------|----------|
| RF | 1.24 | 0.44 | 0.01 | 0.95 |
| SSN | 0.99 | 0.81 | 0.05 | 0.76 |

Figure comparing thermal peak estimates from (a) SSN, (b) RF, and (c) the difference between RF and SSN.

14. Discussion section lines 355-357 discussing differences among models in which explanatory variables are important. This to me, is one of the challenges and potential disadvantages to using a multi-model approach. It can result in a muddled inferential

picture and therefore which variables might be important to emphasize to land managers or conservationists that are concerned about habitat restoration actions for stream temperature. For the multi-model approach to offer significant benefits, it seemingly should provide more robust and improved predictive performance, while caution is exercised regarding the interpretation of variables affecting the response metric.

We agree and will include a discussion of the pros/cons of the different models. In the Methods, we will remind that the multi-model approach is frequently used to account for uncertainties in studies of climate change impact and in hydrological forecasting systems. This approach was borrowed from the modellers community and carried out thereafter for predicting flow characteristics in ungauged basins (see Razavi and Coulibaly (2016), doi:10.1080/02626667.2016.1154558 for a recent application). More reliable predictions at ungauged locations are expected by combining single model estimations. In the Discussion, we will specify that whereas the multi-model has the best performance, it lacks the explanatory power and relative simplicity of the other approaches. Another benefit of the multi-model approach is that by leveraging multiple approaches, it can compensate for errors particular to individual models.

15. Discussion section, lines 361-370 discussing the use of air temperature as a proxy for stream temperatures. While the use of air temperatures was common one or two decades ago, it's become much less common in recent years with the broad availability of stream temperature datasets and interpolated map scenarios like the author's have created here. Towards that end, it would be useful to discuss how your datasets will be made available to others so they can benefit from them. The large temperature observation dataset would be of great utility to researchers conducting thermal regime research, whereas the thermal peak scenarios could be used by aquatic ecologists in France developing species distribution models or assessing vulnerability to climate change.

We agree that air temperature is not as common as it once was, but would argue that it is still in use because datasets like the one presented here are still relatively rare. However, we will reduce some of this stronger language throughout. We will further include some additional text to discuss how the dataset can be made available and used by ecologists in France and scientists more broadly. We note that part of the database (approximately 600 stations) is publicly available from Naïades, which we now include this in the Methods.

The majority of other data is sparse, typically only with summer information. We have created a website to be able to share this data, and will include its information in our revisions (thermie_rivieres.fr).

---

## Author Comment (AC2)

Journal: HESS
Title: The thermal peak: A simple stream temperature metric at regional scale
Author(s): Aurélien Beaufort et al.
MS No.: hess-2021-218
MS Type: Research article
**Responses to Referees**

**Anonymous Referee #2:**

We thank Referee #2 for their detailed review of our manuscript. We have broken out your individual comments, which are numbered, and responded to each accordingly in blue. We hope that our comments address and clarify any issues or concerns that they may have.

**Comments**

In Beaufort et al. the authors compiled stream temperature data for the entirety of France and Corsica, calculated an ecologically relevant summarizing metric ("the thermal peak"), and compared predictive models and an air proxy to determine the best model structure and predictor variables for extrapolating the thermal peak metric to all rivers in France/Corsica. Although the work done within is a valuable contribution, the manuscript would be improved by (1) altering the climate-change framing of the paper, (2) including a specific question or hypothesis, and (3) increasing specificity and clarity of the presentation. Regarding the first point, the paper frames the research as important for understanding stream thermal regimes under climate change. However, the authors calculate a metric which does not help us understand how climate change alters stream temperatures. This is a major issue. Additionally, there lacks a clear question or hypothesis in the introduction – another major issue.

We agree that the manuscript would be improved by reducing the focus on climate change in-of-itself, and honing the Introductory rationale towards the need to improved stream temperature datasets (our contribution) while acknowledging that climate change as the motivating driver. We also agree that ending the Introduction with a question and hypothesis will create a stronger rationale for our approach. Please see below for our more specific changes that will be made.

1. Comments regarding emphasis on climate change:

    The introduction leads me to think that the authors would be explaining how stream temperature is changing in France/Corsica as a result of climate change.

For example, see: Line 36-37: "However, the magnitude and direction of these expected changes will depend strongly on patterns of stream temperature change"

Line 38: "Hence it is critical to describe and analyze the spatiotemporal variability of river thermal regimes."

Line 71-74: "Indeed, stream temperature metrics that focus on extreme periods […] are likely adequate to understand trends of increasing pressures on aquatic ecosystems."

And in the Methods: Line 102-103: "To address ecologically meaningful temperature metrics under climate change, we focused on the two hottest stream temperature months, …" However, the metric the authors calculate does not describe temporal stream temperature changes. The described calculation of the thermal peak is an average of peak temperature across years for each logging station. The metric is thus only useful for understanding the spatial pattern of the thermal peak across France/Corsica for the years in this study. Although the compiled database itself may be a useful source for studies on climate change, the thermal peak metric calculated in this study does not describe changes in stream temperature through time.

Indeed, the authors even admit to this shortcoming in the discussion: Line 315-316: "The downside of the current approach is that it remains based on interannual metrics. Indeed, the non-concomitance of the time series does not allow us to compare extreme years (hot vs. cold)." More emphasis should be put on the thermal peak being a good comparison metric for rivers across space and may help managers understand which rivers currently exceed thermal tolerances of important biota.

These are all good points and we will alter the Introduction and Discussion accordingly. We will focus more on the main point of the paper throughout, which is the creation of a homogenous stream temperature database and the development of a simple metric to understand spatial variation in extremes, which of course, are a result of increasing air temperature due to climate change. Hence, we will remove/edit most of the referenced text here so that the objective is clear. We also will add text to lead to the eventual comparison of stream temperature records/extrapolation with air temperature, as our dataset offers an important improvement over often-used air-temperature proxies.

2. Comments regarding lack of question/hypothesis:

a. An objective is stated in the introduction (Line 73-74) but no specific question, hypothesis or prediction is explained. This left me wondering throughout the paper what the purpose of the research was. What questions motivated this research? What did you expect to learn from research at hand? What do you expect the thermal peak tell us about rivers in France/Corsica?

We agree that these additions will improve the rationale for this work, and we will include the following question and hypothesis in the Introduction. What are the spatial patterns of stream temperature extremes in France and their drivers, and are these patterns consistent across modeling approaches? We hypothesized that spatial patterns would be consistent, whereas the drivers would depend on the modeling approach used. We also hypothesized that stream size, air temperature, and groundwater contributions would emerge as important regardless of approach. Overall, we maintain our objective to create a harmonized database of daily stream temperature in France over the recent decade of warming as an important result of this work. We will demonstrate the importance of this result now by focusing the Introduction more on the current lack of such databases in the literature.

To evaluate our hypothesis, we will include an additional result showing the direction and magnitude of variable effects on the thermal peak (i.e., not just the variable importance for each model) that will allow us to evaluate their influence. To illustrate an example result here, we will compare the spatial variability of the thermal peak in relation to air temperature and stream size (see below).

[Figure]

Figure showing the reach distributions of thermal peaks (y-axis) for random-forest-modeled stream temperature (red) and air temperature (blue) as a function of stream order (x-axis). We can see that whereas thermal peaks based on air temperature are insensitive to stream order, those based on stream temperature rapidly increase with stream order. Importantly, below Strahler order 5, stream temperature thermal peaks are less than air temperature, but this trend reverses for larger rivers. Hence, we can see that stream thermal peaks are more sensitive to network location than to air temperature.

b. Section 4.3 of the discussion aims to attach meaning to the most important explanatory variables across each model. There was no hypothesis about which explanatory variables were thought to be the most important predictors, therefore it seems the authors are fishing for an explanation without having a clear question, hypothesis or prediction about these explanatory variables. Since the author's purpose for making these models was to predict the thermal peak across all rivers of France/Corsica, it is inappropriate to interpret the explanatory variables in this way.

We agree that variable selection was not clear in original text. We will improve our rationale for the explanatory variables used by being more explicit in our questions and hypothesis (see our response above), and by including specific hypothesized effects for each variable in Table 2 (see below). Indeed, each variable has a precedent for controlling stream temperature in the literature and was evaluated for collinearity (see correlation matrix that will now be in the Supplementary Material) before application in the models. We also now include more text in the Methods providing rationale for our variable choices, where some of the variables have substantial literature support, some have more recent, but less tested support (i.e., effects of ponds and weirs), and some we chose to test in this paper (i.e., concavity index and hydrological regime). These clarifications in our rationale for variable selection allow us to more readily attach meaning to their relative importance and effect direction (on $T_p$) in the Discussion.

**Table 2.** List of explanatory variables used in models.

| Category | Variable | Notation | Source | Possible effect |
|---|---|---|---|---|
| Climate | Mean annual precipitation (2009 –2017) [mm] | $P_{annual}$ | SAFRAN | Contrast between climatic regimes (Moore et al, 2013) |
| | Mean summer precipitation, July–August (2009 –2017) [mm] | $P_{summer}$ | SAFRAN | Influence of heat budget Caissie, 2006 |
| | Mean annual snow accumulation (2009 –2017) [mm] | $S_{annual}$ | | Heat budget, meltwater influence, Caissie, 2006, Webb et al, 2008 |
| | Mean summer air temperature, July–August (2009 –2017) [°C] | $T_{summer}$ | SAFRAN | Positive effect related to heat budget, relative Moore et al, 2013 (index of the thermal summer climate) |
| | | | SAFRAN | |
| Hydrology | Mean annual specific discharges [L s$^{-1}$ km$^{-2}$] | $Q_{mean}$ | | Thermal capacity influence, Caissie, 2006 |
| | Mean monthly minimum specific discharge with a return period of 5 years* [L s$^{-1}$ km$^{-2}$] | $q_{min}$ | RHT | Proxy of base flow index, Chang and Psaris, 2013 |
| | Concavity index† [-] | CI | RHT | Proxy of water storage in the catchment (snow or groundwater), tested in this paper |
| | Hydrological regime‡ [-] | HR | RHT | Contrast between hydrological regimes, tested in this paper |
| | | | RHT | |
| Catchment characteristics | Mean catchment elevation [m] | elev | RHT | Negative effect given the relation with air temperature (Isaak and Hubert, 2001) |
| | Drainage area [km²] | area | RHT | Proxy for width-depth ratio of streams (Hrachowitz et al, 2010), and thermal capacity (Imholt et al, 2013) |
| | Mean streams slope over the catchment [m km$^{-1}$] | slope | RHT | Affect river hydraulics and thus thermal advection and exposure time to incoming radiation (Daigle et al, 2010) |
| | Riparian vegetation cover ratio in 10 meters buffer (%)** | veg | SYRAH | Negative effect, as decrease exposure to diurnal radiation (Moore et al, 2005) |
| | Linear weir density upstream of stations (# km$^{-1}$)** | | | Potentially heating effect (Chandesris et al, 2019) |
| | Areal weir density upstream of stations (# km$^{-2}$)** | weirs | SYRAH | Potentially heating effect (Chandesris et al, 2019) |
| | Pond cover ratio upstream of stations (%)** | weir area | SYRAH | Potentially heating effect when ponds and shallow reservoirs release warm water from overflow (Seyedhasemi et al, 2021) |
| | | ponds | SYRAH | |

3. Comments regarding improvements on clarity and specificity:

a. Line 88, sentence starting with "Hence, our main challenge was to pool...": This sentence was difficult to understand. Improve with a more thorough explanation, possibly breaking up this sentence into multiple.

We agree this is confusing. We will clarify that the challenge was coalescing and harmonizing all the disparate data sources.

b. Line 99, "data were ... averaged into mean daily stream temperature data". Do you mean "daily mean" here? It is unclear if it is meant to be the mean daily temperature (averaged across years for a particular day) or the daily mean temperature (average

temperature of a specific date).

Indeed, we meant the latter: that the means of hourly data were taken to achieve daily mean temperatures. We will be more clear here.

c. Line 100, "1) hourly Tw anomalies". If Tw is defined as the daily mean temperature, how can you see hourly anomalies if there is one mean temperature per day? May need to define another variable to use here.

Thank you for catching this. We will change this to "hourly stream temperature anomalies" to avoid conflating Tw with the hourly data.

d. Line 101, "Tair": Need to define this variable because it is the first time it is used.

Fixed.

e. Section 2.2, first paragraph. Why hold back on defining the "simple metric" within the first sentence? It is odd to hold back on defining the metric and then reveal the big mystery of what the metric is.

We do not quite follow this comment, but we will add "…, the thermal peak." In the first sentence to clarify.

f. Section 2.2, second paragraph. $\overline{T_{w,30}}$ is defined as the "mean temperature of the 30 hottest consecutive days of each year". However, the data have already be constrained to July and August. How can you be sure that the 30 hottest consecutive days occur completely within the window between July 1 and August 31? In the last sentence you conclude that the hottest day of the year always occurred between July 28 and August 30. This implies that for the temperature time-series in which August 30th is the hottest day of the year, the 30 hottest consecutive days would include days in September. So, $\overline{T_{w,30}}$ should be re-defined in the paper as the mean temperature of the 30 hottest consecutive days in July and August. Be more specific about defining $\overline{T_{w,30}}$.

We now include in the Supplementary Material that our analysis shows that July and August are generally the hottest months of the year (see below for analysis on stations with annual data). Importantly, many of the stations, particularly those operated by fishing agencies, only have July and August data, which we will precise in the Methods. We agree that we should be more specific and clear in this section and will make the appropriate changes to the text noting that analyses are constrained to July and August. To support our approach, we will also include a sensitivity analysis on the sites with

annual data where we compare the $T_p$ of the annual time series with the $T_p$ limited to July and August.

[Figure]

hottest month of the year across reaches

g. Line 164, "SAFRAN data". Define what this data is.

We will include a definition in text that explains the following: (Système d'analyse fournissant des renseignements atmosphériques à la neige) is a mesoscale atmospheric analysis system for surface variables with reanalysis data at hourly time steps using ground data observations. Originally intended for mountainous areas, it was later extended to cover France. The detailed description can be found in Durand et al. (1993, 2009).

h. Line 198, "… all possible variables." Very broad. More specificity would be helpful, such as "all possible variables characterized in Table 2."

Agreed.

i. Line 236-238, "This bias … in the regressions." These sentences are difficult to understand. Be more specific to improve clarity. The ending of one sentence is "…with only one year of observation.", and the next sentence begins, "In contrast, when there is only one year available…". These aren't contrasting clauses, thus making it difficult to understand what is being contrasted.

Agreed, we will improve clarity here.

j. Sentence starting on line 260, ending line 263, "The two most relevant…". Break into 2 sentences to improve clarity.

Agreed, we will improve clarity here.

k. Line 281: use of 14°C. Explain why 14°C important and why this value chosen as a cut-off point.

This was predominately chosen for ease of visualization. We will explain this in the figure and in text.

l. Figure 6C. Figure 6c shows probability distributions for extrapolated thermal peak. Firstly, the bins are not of equal size: the first bin (labelled <14) includes all temperatures between 6.3 (minimum temperature reported on line 277) and 14, a range of 7.7 degrees; the second through fifth bin have ranges of 3 degrees; and the final bin includes temperatures from 22 to 27, a range of 5 degrees. The 14- and 22-degree cutoff is not explained and thus seems arbitrary. Secondly, since temperature is continuous, it shouldn't be binned in this way, unless for some specific reason. Continuous distributions should be portrayed as "probability density functions". Each model would then have a continuous distribution, and each can be plotted on the same graph for comparison.

Originally, it was made this way to correspond to the cut-bins in Figure 5, which were made to best visualize the temperature distributions. We will made the requested edits to the figure.

m. Line 301, first sentence in discussion. Be more specific about the timeframe of the dataset: "the largest, regional summertime stream temperature datasets"

Agreed, we will include the timeframe of 2008–2018.

n. Line 326. Which "metrics based on observations", specifically, are you comparing to estimates of Tp?

We will more specifically refer to the fact the thermal peak calculated with observed data containing gaps $T_{p,obs}$ (i.e., without the climate correction) has larger biases than thermal peaks calculated with observations that are climate corrected $T_{p,fill}$

o. Line 328. Further explain what is meant by "when applied at scale". How does that relate to figure 6B?

Good point, removed.

p. Line 352: "Higher minimum flows" would make more sense than "Larger minimum flows"

Agreed and changed.

q. Line 354: Change "greater" to "more". "Great" implies a quality of shade and not a quantity. "More" implies a quantity.

Agreed and changed.

4. Technical Corrections

a. Line 21: Use of "However" does not belong because this sentence doesn't relate to previous

b. Line 27: "…, and decomposition rates, and dictates animal … "

c. Line 33: "and levels" is repetitious with flows

d. Line 96: "… were previously…" change to "have been". "were previously" implies that a different study excluded these data. If that is the case, cite that study.

e. Line 155: "is not clear" should be "was not clear"

f. Line 221, equation 7: regression coefficient "a" is in the equation twice and "c" is missing.

g. Line 229: "cross-validation this 100 times"

h. Line 321: insert "between" between "correlation" and "Tair"

i. Line 325: remove "particularly"

j. Line 326: remove "a robust interannual metric". Only need to say "... sufficient to estimate Tp"

k. Line 329: change "to" to "of" in "climate corrections to temperature metrics"

l. Line 352: change "following" to "followed"

m. Line 373: change "representativeness" to "representation"

All suggested technical corrections will be made.

---

## Referee Report (RR1)

Beaufort et al. Review of Revised Manuscript

While improvements were made in the revised version of Beaufort et al., the manuscript does not adequately describe the utility of the "thermal peak" metric (comment 1). The addition of a question and hypotheses were improvements but were tacked on the end of the introduction with little relevance to the presented problems and no rational as to how the authors came up with them (comment 2). Additionally, the manuscript lacks clarity, uses imprecise language, and has many technical errors that need to be addressed (comment 3).

Comment 1

(A) The authors are continuing to frame their metric as useful for understanding climate change, see lines 70-75:

"However, for management purposes in the context of climate change, annual stream temperature patterns may be unnecessary. Indeed, stream temperature metrics that focus on extreme periods (e.g. summer) are likely adequate **to understand trends of increasing pressures on aquatic ecosystems** [*emphasis added*].

**To that end** [*emphasis added*], our objectives are twofold: 1) to create of a harmonious stream temperature database for France, and 2) to develop empirical statistical models to predict a simple, ecologically relevant stream thermal metric that captures the magnitude of the stream temperature extremes at the regional scale."

As stated in my first review, the metric the authors are calculating in the study does not lead to better understanding of trends of increasing pressures on aquatic ecosystems (as promised in line 72). The "thermal peak" metric removes all temporal variation and thus any potential for understanding how summertime temperature extremes are trending due to climate change because the metric takes the mean across years. I would encourage the authors to really consider what this metric *can* and *cannot* tell us about stream temperature regimes. Again, I don't deny that the thermal peak metric is useful for understanding spatial variation in temperature extremes and this metric may be useful for management decisions. However, I disagree with the author's statement in their response to review that the spatial variation in stream temperature extremes are a result of increasing air temperature due to climate change. Imagine a scenario where the climate, and thus air temperature, has not changed, we would still expect there to be spatial variation in annual maximum temperatures across different streams and even in different locations, longitudinally, in the same stream (See Fullerton et al. 2015 for examples of spatial variation in stream channel temperature). (Referenced comment: "We will focus more on the main point of the paper throughout, which is the creation of a homogenous stream temperature database and the development of a simple metric to understand **spatial variation in extremes, which of course, are a result of increasing air temperature due to climate change** [*emphasis added*]**.**")

(B) Lines 383-385: "…we employed a combined empirical approach that allowed us to identify, at the regional scale, **a map of summer stream temperature maxima with important implications for aquatic species distributions under climate change** [*emphasis added*]**.**"

What exactly are these important implications? A thorough explanation after this sentence or an example of how the authors imagine the map being used would provide much needed clarity on this point.

(C) Lines 393-394: "This map can presently be used to **predict and manage future cold-water habitat streams** [*emphasis added*], with potential for regular multi-annual updates."

I don't see how this map of summertime maximum stream temperature averaged over 2009-2017 can be used to predict where future cold-water habitat streams will be. I can make assumptions about what the authors are getting at here, but that is left unsaid. I am assuming the authors mean: This map can be used as an initial benchmark for understanding the summertime temperature extremes of streams across France and upon update with new data, we can determine where the thermal peaks are increasing the greatest, thus allowing managers to focus their efforts are the streams with greatest increase in thermal peak. This is possibly not what the authors meant here, but either way, the reader shouldn't have to guess as to what the authors are getting at. I do see the utility of the database, the presented maps, and their analysis of the different models, however the utility is not clearly presented in the manuscript.

Comment 2

In the Introduction, the authors lay out multiple problems: (1) lack of long-term stream temperature data, which is often spatiotemporally uncoordinated (see L 44-46), (2) stream temperature data are often supported or replaced by air temperature proxies (see L 46-47), and (3) other statistical gap-filling methods require a lot of data, which may be unnecessary for management purposes in the context of climate change (see L 66-67, 70-71). Later, the authors describe their question as "…what are the spatial patterns of stream temperature extremes in France and their drivers, and are these patterns consistent across modeling approaches?" (L 79-81). Previous to this question, there has been little mention of the drivers of stream temperature patterns nor a mention of the different modeling approaches in question. The Introduction would be greatly improved if the problems were more specific to their question. Additionally, the authors did not explain the rational for the listed hypotheses. Why would they expect the spatial patterns to be consistent across modeling approaches and the drivers predicted by the models to be different? The lack of rational for these hypotheses and the fact that their results exactly confirm this hypothesis leads me to think the authors retroactively came up with this hypothesis after they got the results they did. Although this may be an inaccurate assessment by me, the way the hypotheses are presented make it seem so.

The second hypothesis lists "stream size, air temperature, and groundwater contributions" as important to the thermal peak metric (L 82-83, 264-265), however, the manuscript uses drainage area "area" and mean monthly minimum specific discharge "$q_{min}$" as metrics for stream size and groundwater contributions, respectively. "Stream size" is a somewhat general description and can be described using many different metrics, thus use "drainage area" in hypotheses because that is the specific metric being used. In section 2.4.1, $q_{min}$ is said to describe the low-flow regime of the site and later in the results this metric is used to describe groundwater contributions. The manuscript does not provide citations or rational as to how $q_{min}$ is a metric of groundwater contributions in streams nor does the writing

describe that they are using $q_{min}$ as a metric for groundwater contributions previous to results, thus the reader must make assumptions after the fact. Again, this could be solved by using "mean monthly min. discharge" in the hypotheses or a description of why the authors use $q_{min}$ as a metric for groundwater contributions.

Comment 3

Additionally, there are numerous technical corrections, from erroneous inclusions/exclusions of words (e.g., L 73 "create  a harmonious", L 210 "we did [not] seek", and others), to mixed verb tense (e.g., L 104-115 use of "have" instead of "had", L 337-339, L374-376, etc.), spelling errors (e.g., L 340 "mulit-model", L 355 "wateshed", and others), and a general lack of precise language, e.g. (list not exhaustive):

- (L 8) "…data are decentralized…": what does "decentralized" mean in this context?
- (L 32-33) "… with different metrics to quantify the biological or ecological importance of each component": unclear what the authors are talking about here
- (L34) "…these metrics…": do you mean components here? The list in the previous sentence was "components" of thermal regimes
- (L 75) "… metric that captures the magnitude of the stream temperature extremes…": the phrase "stream temperature extremes" includes a maximum **and** a minimum, and the metric presented only addresses the annual stream temperature maximum.
- (L 94) "…harmonize all the disparate data sources.": what exactly is data "harmonization"?
- (L 98) "…stations without seasonal dynamics have been excluded…": What was the determining criteria for "no seasonal dynamics"?
- Table 2. Definition of "n" as count of days. These numbers seem like they are the numbers of sites used, not the number of days.
- Table 3. In "Hypothesized effect" of CI and HR. These are not hypothesized effects, just definition of terms.
- Figure 7. It is unclear why (a) and (b) are presented together here. The difference in models is shown in (a) and the difference in drainage area for one model is shown in (b). Why was the RF model chosen to display in (b) and not the other models?

As well as areas that could use an improvement to organization:

- (L 385) "We note here…": this sentence seems out of place here and is lacking a rational for the method that you *did* decide on. After reading this sentence I'm left wondering, well why didn't you just a simple mean of August data?
- (L 487) "Groundwater and shading proxies …": this sentence also seems out of place.

---

## Referee Report (RR2)

Line 42-43: …,  reaches with strong local controls …and  reaches with large environmental heterogeneity …

- Unnecessary to include the "in" in each element of the list

Line 45: "Hence, while air temperature has clear utility for understanding stream temperature, …"

- The entire paragraph up until this point was spent convincing the reader that air temperature was a poor surrogate for stream temperature, however, in this sentence they are saying air temperature it has clear utility for understanding stream temperature. I think the author's point about air temperature having clear utility should be clarified because it does not make sense given what was explained in the previous sentences of this paragraph.

Line 51: "As streams  surface area increases…"

- Referring the stream size is ambiguous here because I am left to wonder if you are referring the stream order or to the stream's width or volume. I believe it is only necessary to talk about the stream's surface area in this sentence.

Line 69: "…countries lack  the …"

Line 72-78: "…our objectives  -> were twofold …"; capture -> captured; define -> defined; term -> termed; estimate -> estimated; test -> tested

- Verbs should be past tense

Line 75: extremes -> maximums

- As suggested in my last review "stream temperature extremes" should be "stream temperature maximums" because this paper is only talking about annual *maximum* temperatures and not minimums. I noticed the terminology had been changed from "temperature extreme" to "temperature maximum" in other areas of the paper, I assume it was just overlooked here.

Line 91: time -> time

Line 186: climate -> climactic, hydrology -> hydrological

- Use the adjective version of the words because they are describing categories

Line 380: demonstrate -> demonstrated

Line 387-388: alleviate -> alleviated; demonstrate -> demonstrated

Line 449-451: I don't see how the second sentence (starting w/ "The importance of") is different from the first. In both sentences the authors are saying that the relevance of mean summer air temperature is consistent with other studies. The Moore cites could be added to the first and the second sentence deleted.

Lines 455-456 & 458: "water travel times" & "residence time"

- Terminology changed from "travel time" to "residence time". I would suggest choosing one and using the one term in both places for clarity.

---

## Author Response (AR2)

Journal: HESS
Title: The thermal peak: A simple stream temperature metric at regional scale
Author(s): Aurélien Beaufort et al.
MS No.: hess-2021-218
MS Type: Research article
**Author Response**

Dear Editor,

You will find the new version of our manuscript.

We would like to thank the reviewers for their very constructive and useful comments and suggestions. This has helped us a lot to improve our manuscript, which I hope will be now suitable for publication in HESS.

Best regards,

Florentina Moatar and co-authors

---

## Author Response (AR3)

Journal: HESS
Title: The thermal peak: A simple stream temperature metric at regional scale
Author(s): Aurélien Beaufort et al.
MS No.: hess-2021-218
MS Type: Research article

**Responses to Referees**

**Anonymous Referee #2:**

We thank Referee #2 for their detailed review of our manuscript. We have broken out your individual comments, which are numbered, and responded to each accordingly in blue. We hope that our comments address and clarify any issues or concerns that they may have.

**Comments**

I was disappointed to find that the authors did not respond with an itemized response to the last review and instead only provided a track-changed document. I found this extremely disrespectful of my time as a volunteer reviewer. Minor changes are suggested, see attached.

We apologize for any time we may have wasted for the reviewer and thank them again for their effort.

1.  Line 42-43: …, in reaches with strong local controls …and in reaches with large environmental heterogeneity … Unnecessary to include the "in" in each element of the list

    Sentence revised accordingly.

2.  Line 45: "Hence, while air temperature has clear utility for understanding stream temperature, …"

    The entire paragraph up until this point was spent convincing the reader that air temperature was a poor surrogate for stream temperature, however, in this sentence they are saying air temperature it has clear utility for understanding stream temperature. I think the author's point about air temperature having clear utility should be clarified because it does not make sense given what was explained in the previous sentences of this paragraph.

    Sentence revised accordingly.

3.  Line 51: "As streams increase in size, their surface area increases…"

Referring the stream size is ambiguous here because I am left to wonder if you are referring the stream order or to the stream's width or volume. I believe it is only necessary to talk about the stream's surface area in this sentence.

Sentence revised accordingly.

4. Line 69: "…countries lack  the …"

   Sentence revised accordingly.

5. Line 72-78: "…our objectives are -> were twofold …"; capture -> captured; define -> defined; term -> termed; estimate -> estimated; test -> tested

   Verbs should be past tense

   Revised accordingly.

6. Line 75: extremes -> maximums

   As suggested in my last review "stream temperature extremes" should be "stream temperature maximums" because this paper is only talking about annual maximum temperatures and not minimums. I noticed the terminology had been changed from "temperature extreme" to "temperature maximum" in other areas of the paper, I assume it was just overlooked here.

   Revised accordingly.

7. Line 91: times -> time

   Revised accordingly.

8. Line 186: climate -> climactic, hydrology -> hydrological

   Use the adjective version of the words because they are describing categories

   Revised accordingly (with "hydrologic" instead of "hydrological", and "climatic" instead of "climactic", as reviewer suggested).

9. Line 380: demonstrate -> demonstrated

   Revised accordingly.

10. Line 449-451: I don't see how the second sentence (starting w/ "The importance of") is different from the first. In both sentences the authors are saying that the relevance of mean summer air temperature is consistent with other studies. The Moore cites could be added to the first and the second sentence deleted.

    Revised accordingly.

11. Lines 455-456 & 458: "water travel times" & "residence time"

Terminology changed from "travel time" to "residence time". I would suggest choosing one and using the one term in both places for clarity

Revised accordingly.

Journal: HESS
Title: The thermal peak: A simple stream temperature metric at regional scale
Author(s): Aurélien Beaufort et al.
MS No.: hess-2021-218
MS Type: Research article

**Responses to Referees**

**Anonymous Referee #3:**

We thank Referee #3 for their detailed review of our manuscript. We have broken out your individual comments, which are numbered, and responded to each accordingly in blue. We hope that our comments address and clarify any issues or concerns that they may have.

**Comments**

I generally agree with the previous two reviewers in that it is a worthwhile study and dataset. I would recommend the paper to be accepted for publication subject to these few minor revisions. Overall, the authors took into account the vast majority of the reviewers' comments. I pointed to comments by R1 and R2, which have not been addressed (or if they were, it was not obvious). Line numbers refer to the revised version of the manuscript.

1. Lines 36-39: Add a mention of water quality studies too (one reference would suffice).
   We inserted line 27 van vliet and Zwolsman work on summer droughts on the water quality of the Meuse river
   "Water temperature extremes, like summer maxima, are particularly strong drivers of aquatic biota behavior,habitat selection (Carlson et al., 2007; Xu et al., 2010), and water quality (van Vliet and Zwolsman, 2008)."

2. Lines 45-46: Add reference to back up statement.

References are provided throughout the paragraph to support this paragraph-concluding sentence. We also modified this sentence a bit based on the other Referee comment.

3. Line 98: Space missing after parenthesis in "power thermal effluent)were excluded".
Sentence revised accordingly.

4. Lines 99-105: Clarify where the daily mean temperature (Tair) data comes from. Note that R2 made a comment asking to clarify what was meant by "mean daily stream temperature", but it is not obvious if that comment has been addressed (R2 review, comment 3.b re line 99 of original manuscript).
The following text was removed from lines 179–183 and added on line 106: $T_{air}$ was provided by the 8 km gridded SAFRAN (Système d'Analyse Fournissant des Renseignements Atmosphériques à la Neige) atmospheric reanalysis data released by Meteo-France over 2009-2017 period (Vidal et al., 2010). It was extracted from SAFRAN meshes overlapping the station location. Mean daily stream temperature was indeed clarified in the previous version on Lines 97–103.

5. Line 113: Clarify "oriented hydrographic network" (do you mean flow direction information are built-in?)
We used common-language for hydrographic networks, but for clarity have also now added the parenthetical: "(i.e., built-in upstream-downstream dependencies)" on L118.

6. Lines 133-138: Clarify paragraph:
R1 commented re section 2.2 (Line 118 in original version). R1 suggested the metric could have been much simpler. R2 also commented on that section, specifically the second paragraph about the definition of the metric. The authors edited that section to clarify the metric, and added a paragraph to explain why they designed it as it is (lines 133-138 in new version). I would say the addressed R1 and R2 comments. However, I find that new paragraph is not very clear. In particular, it is hard to visualise what this paragraph means in practical terms were you to collate data and calculate the metric. I would suggest revising that paragraph.
Indeed, we revised this based on original Referee comments. It is not easy to revise this paragraph as requested by you (Referee 3) without some more concrete or specific suggestions. We have elected to leave it as is because the metric is actually defined on

liens 128–137, not in this paragraph. This paragraph simply serves to provide rationale for not *a priori* using the maximum of say, August, data.

7. Line 181: replace "fit" with "fitted".

   Either are fine, technically; US speakers tend to use "fit".

8. Table 3: In Reference, just repeat Sauquet et al (2008) and Sauquet and Catalogne (2011) for HR and CI rather than send reader back to text.

   The Reference in this case is referring to the Hypothesized Effect, not the variable, itself, so the References are correctly stated. Left as is.

9. Line 203: spell out SYRAH-CE.

   Revised accordingly.

10. Lines 210-237, Sections 2.4.3 and 2.4.4: Was text revised to address R1's comment?

    R1 commented on sections 2.4.2 to 2.4.4 (his comment refers to "Methods, lines 197-215") requesting that more details should be included "that a knowledgeable reader could replicate the analysis". I can see that section 2.4.2 was edited accordingly and is fine, but sections 2.4.3 and 2.4.4 are the same as in the original version; I am not very familiar with ANN or random forest so maybe these sections are detailed enough but they look rather sparse, especially 2.4.4. Could the authros confirmed the text has been revised and left unedited on purpose?

    We confirm that we revised this text, but we have provided additional detail as requested. For ANN and RF, there are standard methods that we follow and already cite, and it is not within the scope of this paper to describe their inner-workings in their entirety.

11. Lines 253 and 265 the section number is 2.4.4 for both; it should be 2.4.6 and 2.4.7

    Revised accordingly.

12. Line 447 section 4.3: see R2 review comment 2.b; could you re-check that comment and confirm you addressed it, and how, because it is not immediately obvious (I appreciate the authors amended the intro and added hypotheses, thus addressing R2 comment 2.a so possibly addressed 2.b implicitly).

    It is not clear what is meant by comment 2b from the R2 review…did the Referee mean comment 1b? Regardless, we confirm that we addressed all comments from that reviewer, including formalizing hypotheses and providing support for $q_{min}$ and area as predictors.

13. Line 463: R2 suggested to change "greater" with "more" but text was not amended; confirm it is intentional.

Revised.

14. Line 437: correct sentence "We tested s SSN model was tested".

Revised.

---

## Author Response (AR4)

Dear Editor,

Many thanks for your revision.

We modified P1L30. We also insert the equation for IC in the manuscript and a new Appendix (Appendix A) to describe HR.

We hope that readers could now a better understanding of these two hydrological indicators.

Best regards,

Florentina Moatar
* * *
The remaining two variables, the hydrologic regime (HR; Sauquet et al., 2008) and the concavity index (CI; Catalogne, 2012), are dimensionless and characterize the general hydrology of each site. More specifically, the HR groups sites into one of 12 classes ranging from rainfall-dominated, to transitional, to glacial and snow melt dominated (see Appendix A). These classes generally fall into buffered (i.e., high baseflow) or highly variable (i.e., low baseflow) hydrologic regimes. The CI describes the concavity of the flow duration curve, where values close to 1 indicate low flow variability (e.g., large high storage capacity in aquifer or snow) and values close to 0 indicate high flow variability (e.g., low storage capacity exemplary of Mediterranean systems). The concavity index is computed as follows:

$$IC = \frac{Q_1 - Q_{10}}{Q_{10} - Q_{99}} \tag{6}$$

where $Q_p$ is the daily flow exceed p% of the time derived from the flow duration curve.
* * *
**5. Appendix A: Classification of river flow regime for France**

Sauquet et al. (2008) defined a classification system for river discharge based on the mean monthly runoff pattern (Fig. A1). Using this classification system, they published a map with each river reach assigned to one class along the main river network.

Generally, the uppermost basins located in mountainous areas display glacial and snowmelt-dominated regimes (Group 1–3). The lower the outlet, the lower the contributions of snowmelt to runoff. Group 4 to 6 were defined to be in the "transition" regime, where the seasonal variation in streamflow is affected as much by precipitation timing as by air temperature and topographic influences on snowpack formation and snowmelt timing. In this regime, high flows are typically observed in spring. Group 7 to 12 are rainfall-dominated regimes. These six rainfall-dominated groups differ along an axis of maximum and minimum monthly discharges. For instance, group 7 is characterized by very low flow in summer, reflecting the lack of deep groundwater storages in the catchment. In contrast, Group 12 exhibits nearly uniform flows through most of the year, and these river reaches are typically found where large aquifers moderate flows.

[Figure]

**Figure A1.** Reference dimensionless hydrographs representative of the classification of river flow regime for France (after Sauquet et al., 2008). The y-axis (CM) represent the percent of each month contributing to total annual runoff, equal to monthly runoff (mm) divided by the mean annual runoff (mm).